# Experience-dependent structural plasticity targets dynamic filopodia in regulating dendrite maturation and synaptogenesis

Chengyu Sheng [1], Uzma Javed[1], Mary Gibbs[1], Caixia Long[1], Jun Yin[1], Bo Qin[1] & Quan Yuan [1]

Highly motile dendritic protrusions are hallmarks of developing neurons. These exploratory filopodia sample the environment and initiate contacts with potential synaptic partners. To understand the role for dynamic filopodia in dendrite morphogenesis and experience-dependent structural plasticity, we analyzed dendrite dynamics, synapse formation, and dendrite volume expansion in developing ventral lateral neurons (LNvs) of the *Drosophila* larval visual circuit. Our findings reveal the temporal coordination between heightened dendrite dynamics with synaptogenesis in LNvs and illustrate the strong influence imposed by sensory experience on the prevalence of dendritic filopodia, which regulate the formation of synapses and the expansion of dendritic arbors. Using genetic analyses, we further identified Amphiphysin (Amph), a BAR (Bin/Amphiphysin/Rvs) domain-containing protein as a required component for tuning the dynamic state of LNv dendrites and promoting dendrite maturation. Taken together, our study establishes dynamic filopodia as the key cellular target for experience-dependent regulation of dendrite development.

---

[1] National Institute of Neurological Disorders and Stroke, National Institutes of Health, Bethesda, MD 20892, USA. Correspondence and requests for materials should be addressed to Q.Y. (email: quan.yuan@nih.gov)

Dynamic dendritic filopodia are long, thin, membranous protrusions broadly observed in neurons at early developmental stages[1–4]. The frequency of filopodia formation decreases once high synapse density is established in the mature circuit[5–8]. Synaptic activity mediated by NMDA receptor activation promotes dendritic filopodia formation, while contacts between filopodia and axons are stabilized by local calcium signaling, thus facilitating synaptic partner selection[9,10]. These observations indicate functional connections between exploratory filopodia and synaptogenesis and suggest dendrite dynamics as the potential cellular substrate for activity-induced structural plasticity[8,11,12].

Molecules related to calcium signaling and cellular communications, such as CaMKII, Rho GTPases, and neurexin–neuroligin cell adhesion complexes, have been identified as factors involved in regulating filopodia stability and density[1,4,13–15]. Although the exploratory dendritic filopodia are clearly linked to synapse formation, their roles in overall dendrite morphogenesis and plasticity, as well as cellular and molecular mechanisms regulating their prevalence during dendrite development, remain unclear[16,17]. In addition, activity-dependent mechanisms have strong impacts on dendrite growth and patterning during the establishment of sensory circuits. Yet, how sensory experience influences the early dynamic phase of dendrite development has not been characterized[18].

Genetic analyses in invertebrate systems have revealed important principles governing dendrite morphogenesis and remodeling[17,19]. However, previous studies on fast dendrite dynamics in *Drosophila* were limited to the actin-rich protrusions in the non-synapse-forming sensory dendrites of peripheral sensory neurons[20,21]. Therefore, to understand the intricate relationships among synaptic activity, dendrite dynamics, and dendrite growth using genetic approaches, we establish a new model to study dynamic dendritic filopodia using *Drosophila* ventral lateral neurons (LNvs). LNvs are a group of projection neurons in the larval visual circuit. Their dendritic arbors form synaptic contacts with the Bolwig's nerve (BN), the axonal projections of the larval photoreceptors[22,23]. Importantly, during development, LNv dendrites exhibit experience-dependent homeostatic plasticity, in which the amount of light exposure received by the animal inversely correlate with the total dendritic length of LNvs[22,23]. Although chronic activity alterations produce robust modifications of the dendrite size and the synapse number, cellular mechanisms underlying this phenomenon were unidentified.

Taking advantage of this unique model system, we performed quantitative analysis on LNv dendrite dynamics, synapse formation, and volume expansion during larval development. The temporal profiles of these processes reveal a developmental coordination between heightened dendrite dynamics and synaptogenesis at early stages of development and a sharp transition of the dendrite from dynamic to stable states as the synapse number reaches saturation. In addition, we found that chronic elevations of visual input strongly influence the prevalence of dynamic filopodia, promote dendrite maturation in young neurons and hinder dendrite growth at later stages.

Using transgenic RNAi-based genetic screens, we identified Amphiphysin (Amph) as an important factor for tuning dendrite dynamics through its functions in organizing the postsynaptic compartment. Amph belongs to a family of proteins containing a BAR (Bin/Amph/Rvs) domain, which interacts with lipids and modulates membrane curvature[24]. While the primary function of vertebrate Amph is the recruitment of dynamin in facilitating clathrin-mediated endocytosis, studies clearly show that *Drosophila* Amph is not involved in this process[24–28]. Instead, it was found in the postsynaptic region at the neuromuscular junction

(NMJ), regulating muscle transverse tubule (T-tubule) formation and the localization of postsynaptic proteins[25,26,28]. In this study, we observed severe deficits in synaptogenesis and dendrite development in *amph* loss-of-function mutants, as well as excessive dendrite dynamics accompanied by impaired synaptic transmission in LNvs with an *amph* knockdown, revealing previously unidentified functions of Amph in regulating the stability of the postsynaptic compartment in the fly CNS.

Our studies characterize the synapse-forming dynamic dendritic filopodia in *Drosophila* CNS, and thus establish a genetic model to study molecular mechanisms underlying the developmental regulation of dendrite dynamics and maturation. Through systematic analyses, we determined the temporal profiles of key events associated with developmental regulations of dendritic arbors and establish dendrite maturation as the critical process that shifts dendrites from the dynamic synapse-forming phase to the stable growing phases during dendrite development. Furthermore, we demonstrate that dendritic filopodia act as the key cellular substrate for experience-dependent structural plasticity. These findings reveal cellular and molecular mechanisms governing the process of dendrite maturation and add to the emerging understanding of structural homeostasis.

## Results

**Dynamic filopodia on LNv dendrites revealed by time-lapse 3D imaging.** We use *Drosophila* larval LNvs to perform genetic studies on the dynamic properties of dendritic arbors. The four LNvs in each brain lobe are labeled specifically using the enhancer Gal4 driver Pdf-Gal4 (Supplementary Figure 1a, b). In addition, using the FLP-out technique, we expressed membrane-tagged GFP in single LNvs and performed two-photon time-lapse imaging of the whole dendritic arbor in intact larval brain explants. LNvs differentiate at the late embryonic stage, around 18–22 h after egg laying (AEL), and then start to make synaptic contact with the BN in the larval optic neuropil (LON) at the 1st instar stage, ~24 h AEL (Supplementary Figure 1c)[22,23]. We imaged during developmental time points 48, 72, 96, and 120 h AEL, corresponding to the periods during which LNv dendritic arbors make synaptic contacts, expand their coverage and establish mature connectivity[22,23] (Supplementary Figure 1c).

LNv dendritic arbors have no spines, but contain highly dynamic branches that closely resemble dendritic filopodia described previously in vertebrate neurons[6,7,29]. Numerous fast extensions and retractions were observed within the 10 min imaging window on the single-labeled LNv dendritic arbor (Fig. 1a, b, Supplementary Movie 1). To analyze these complex high-resolution 4D image data sets, we developed a new semi-automatic method. Facilitated by 3D image visualization software, we manually marked all branch tips and used automatic spot tracking functions to produce X, Y, Z coordinates at each time point. We then calculated the number, duration, and length of extensions and retractions for existing and newly appeared branches. Our semi-automated approach provides quantitative descriptions of the dynamic state of the entire dendritic arbor and generates direct visual representations of the dynamic behaviors during the imaging period (Fig. 1a, b, Supplementary Movie 2).

**LNv dendrites change from dynamic to stable during development.** During larval development, we observe a clear change in the dendrite dynamics. Dendritic arbors displayed significantly more dynamic behaviors in young 2nd instar larvae (48–72 h AEL) than in 3rd instar larvae (96–120 h AEL) as measured by percentage of dynamic branches, the number of newly appeared branches and the distance traveled by extensions and retractions, as well as the number of dynamic events (Fig. 1b–d,

Supplementary Movies 3-5). Further analyses on adjacent time points reveal significant differences between samples collected at 72 and 96 h AEL (Fig. 1c, d). Thus, LNv dendrite dynamics are developmentally regulated with the transition from dynamic to stable states in LNv dendrites occurring between 72 to 96 h AEL.

We further characterized the dynamic behaviors of LNv dendritic filopodia by analyzing the number and distribution of

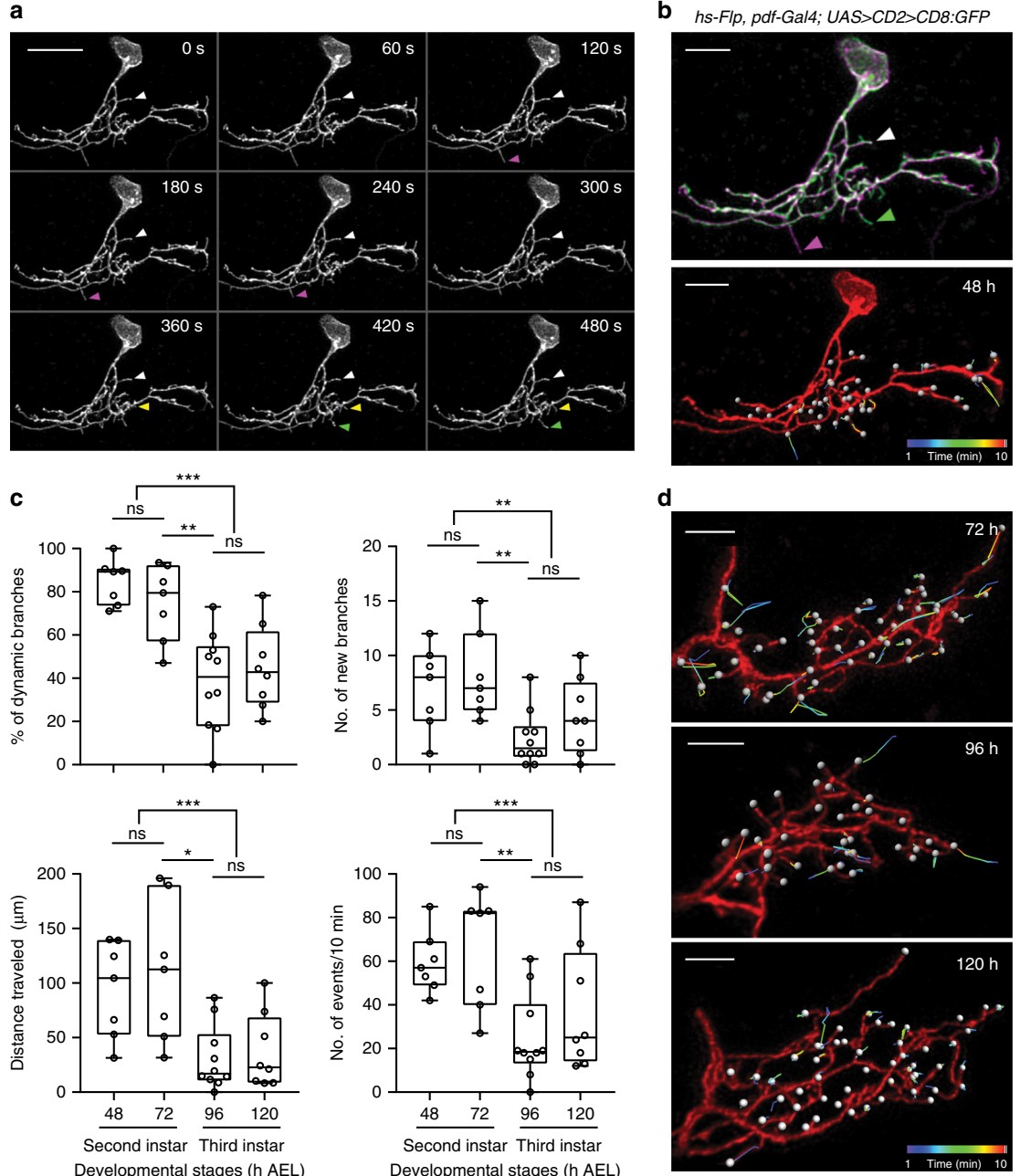

**Fig. 1** Time-lapse 3D imaging of dynamic filopodia on developing LNv dendrites. **a** Examples of extension (green), retraction (magenta), newly appeared (yellow), and stable (white) branches on the dendritic arbor of a single-labeled LNv. Scale bar, 10 μm. A representative Z-projected time series is shown. **b** 4D tracing of branch tips provides accurate representations of dynamic events on the entire dendritic arbor. Top: Overlay of the first (magenta) and last frame (green) of the Z-projected time series. Bottom: The 3D rendering image of tracking trajectories generated by movements of dynamic filopodia. Branch tips are marked by white spots. **c** Decreased dendrite dynamics in mature neurons (3rd instar, 96 and 120 h) as compared to young neurons (2nd instar, 48 and 72 h). Transitions from dynamic to stable state occur between 72 to 96 h AEL. Data are presented as box plot (box, 25–75%; center line, median) overlaid with dot plot (individual data points). $n = 7, 7, 10$, and 8 for 48, 72, 96, and 120 h, respectively. Statistical significance was assessed by one-way ANOVA with Tukey's post hoc test for all time points and two-tailed Student's t-test for grouped comparisons (2nd vs 3rd instar). P-value from Tukey's test for 72 vs 96 h is reported. The percentage of dynamic branches (Top left): ANOVA: $F_{3, 28} = 11.68$, $P < 0.001$; Tukey's: $P = 0.002$; t-test: $t_{30} = 5.834$, $P < 0.001$. The number of newly appeared branches (Top right): ANOVA: $F_{3, 28} = 4.826$, $P = 0.008$; Tukey's: $P = 0.01$; t-test: $t_{30} = 3.553$, $P = 0.001$. The distance traveled by branch tips (Bottom left): Tukey's: $P = 0.004$; t-test: $t_{30} = 4.551$, $P < 0.001$. The total number of events in 10 min (Bottom right): ANOVA: $F_{3, 28} = 5.8$, $P = 0.003$; Tukey's: $P = 0.006$; t-test: $t_{30} = 3.997$, $P < 0.001$. ns, not significant, *$P < 0.05$, **$P < 0.01$, ***$P < 0.001$. **d** Representative 3D rendering images of tracking trajectories generated by movements of dynamic filopodia in single-labeled LNvs collected at 72, 96, and 120 h. Scale bar, 5 μm

extension and retraction events with respect to their duration and distance traveled. Dendritic filopodia are highly motile, switching between extension and retraction or periods of transient stability within minutes. The majority of filopodial extensions and retractions are less than 2 μm in length and last less than 2 min. Interestingly, compared to the mature neurons, we observed significantly more extensions and retractions that travel further than 2 μm or last longer than 2 min in young neurons as compared to mature neurons (Fig. 2a–d). Furthermore, the average rate of extensions changes over development; in young neurons (72 h AEL), extensions are rapid, averaging 1.113 ± 0.044 μm/min (mean ± SEM, $n = 11$ single-labeled neurons), but are significantly reduced by 96 h AEL to 0.738 ± 0.075 μm/min ($n = 13$ single-labeled neurons) (Fig. 2e). A similar reduction in speed was observed for retractions in LNvs by 96 h AEL (Fig. 2e). These results show that dendrites in mature neurons have fewer dynamic filopodia with lower motility.

Our experimental system monitors all branches on the entire LNv dendritic arbor, allowing us to observe two types of dendritic branches that exhibit different behaviors. By quantifying the persistence of branches based on the cumulative stable time during the 10 min imaging period, we observe that dynamic vs. stable populations of branches are clearly segregated and the ratio of these two distinct populations shifts during the process of neuron maturation (Supplementary Figure 2). In addition, at all developmental stages, we observe no net changes generated by dynamic filopodia in overall dendrite length (Fig. 2f). This result suggests that dynamic filopodia are not a major contributor to dendrite growth for developing LNvs, and the non-exploratory dendritic branches likely provide the main sources of dendrite expansion during development. Our observations closely resemble findings in chick retinal ganglion cells (RGCs) and *Xenopus* optic tectal neurons, where fast dendrite dynamics do not generate changes in dendrite length[6,7,29].

**Temporal coupling of dendrite dynamics with synaptogenesis.** What are the functions of these dynamic filopodia in LNv dendrite development? To answer this question and to investigate the connection among synaptogenesis, dendrite dynamics, and dendrite growth, we established temporal profiles of these events during larval development. LNvs start to make functional synaptic connections with the presynaptic terminals around 24 h AEL[22], but the progression of synaptogenesis on LNv dendrites at later stages was unknown.

To analyze the number of synaptic contacts between LNvs and the BN, we used *Rh5,6-Brp:mCherry*, a transgene containing the presynaptic active zone component Bruchpilot (Brp) tagged with mCherry and driven by the photoreceptor-specific enhancer sequence. The *Brp:mCherry* puncta mark the presynaptic release sites and function as a genetic tool for quantitative measurement of the potential synapses in the fly CNS[30,31]. In this set of experiments, we imaged LNv dendritic arbors labeled by membrane-tagged GFP together with *Rh5,6-Brp:mCherry* at 48, 72, 96, and 120 h AEL (Fig. 3a). From the data set, we first quantified the volume of LNv dendrites using 3D surface reconstruction. Next, we measured the number of presynaptic terminals from photoreceptor axons by quantifying the total number of *Rh5,6-Brp:mCherry* puncta in the larval optical neuropil (LON) region, where the BN makes synaptic contact with LNvs and other target neurons[22]. Lastly, we obtained the number of synaptic contacts between LNv dendrites and the BN by quantifying the *Rh5,6-Brp:mCherry* puncta directly contacting the reconstructed surface of LNv dendrites[32] (Fig. 3b, c, Supplementary Figures 3, 4).

These three parameters exhibit different temporal profiles. The LNv dendrite volume continues to increase until 96 h AEL,

similar to the number of total presynaptic terminals made by the BN in the LON (Fig. 3c). However, the number of synaptic contacts between LNvs and BN reaches saturation at an earlier time point, 72 h AEL, coinciding with the peak of dendrite dynamics (Fig. 3c, d). Therefore, our analyses demonstrate a temporal coupling between the heightened dynamic state of LNv dendrites and synaptogenesis, but not volume expansion from the growth of dendrites. These observations support our model in which young dendritic arbors contain a high percentage of dynamic filopodia that correlate with synaptogenesis. As neurons mature, dendrites make the transition into a stable state that supports continued growth.

Taken together, our analyses reveal that dynamic filopodia on LNv dendrites exhibit morphological, behavioral, and functional similarities to vertebrate dendritic filopodia, and thus establish a useful model of in vivo studies on dendrite dynamics in a genetically tractable system.

**Experience-dependent modification of LNv dendrite dynamics.** Previously, we observed that chronic modifications of environmental light:dark conditions during larval development significantly affected total length of the dendritic arbor in LNvs, demonstrating the powerful influence of visual experience and activity-dependent regulation on dendrite morphogenesis[23]. We went on to investigate whether visual experience can affect LNv dendrite dynamics and synaptogenesis.

By modifying the culture light conditions, we provided either excessive visual input or visual deprivation during larval development and monitored the dynamic behavior of LNv dendrites at 72 and 120 h AEL. LNv dendrites in larvae cultured under constant light conditions (light:light, or LL) showed significant reductions in the number of dynamic branches and newly appeared branches compared to larvae under 12 h light:12 h dark conditions (LD). In contrast, constant darkness (DD) conditions have the opposite effect and elicit more dynamic branches and more new branches (Fig. 4a, b, Supplementary Movies 6, 7, and 8). At the same time, light conditions have no significant impact on the properties of the dendritic filopodia. In all conditions, the majority of filopodial extensions and retractions remain less than 2 μm in distance and less than 2 min in duration (Supplementary Figure 5).

Notably, young LNvs (72 h AEL) cultured in LL exhibit dendrite dynamics comparable to those of mature LNvs (120 h AEL) cultured in LD (Fig. 4c), suggesting that elevated inputs potentially shifted young LNvs to a mature state characterized by reduced numbers of exploratory dendritic filopodia and a consequently decreased capacity for synapse formation. We tested this hypothesis by analyzing the effect of LL conditions on synapse formation and dendritic volume at different developmental stages. As expected, compared to LD conditions, both parameters show a significant reduction in LL at 120 h AEL (Fig. 5). Importantly, LL leads to a premature saturation of the number of synaptic contacts and dendrite volume expansion. The number of synapse contacts remains steady after 48 h AEL while the expansion of dendritic arbors plateaus at 72 h AEL, both of which are advanced compared to LD. In contrast, LL conditions do not produce a reduction of total presynaptic terminals generated by the BN in the LON, with *Rh5,6-Brp:mCherry* puncta showing a similar trend of increasing in both LL and LD, indicating a specific effect of LL on the formation of synaptic contacts between the BN and LNv dendrites (Fig. 5).

These findings are consistent with our model that chronic alterations of synaptic input regulate the capacity for synaptogenesis and dendrite growth by modifying dendrite dynamics. Our observations further strengthen the role of experience-dependent

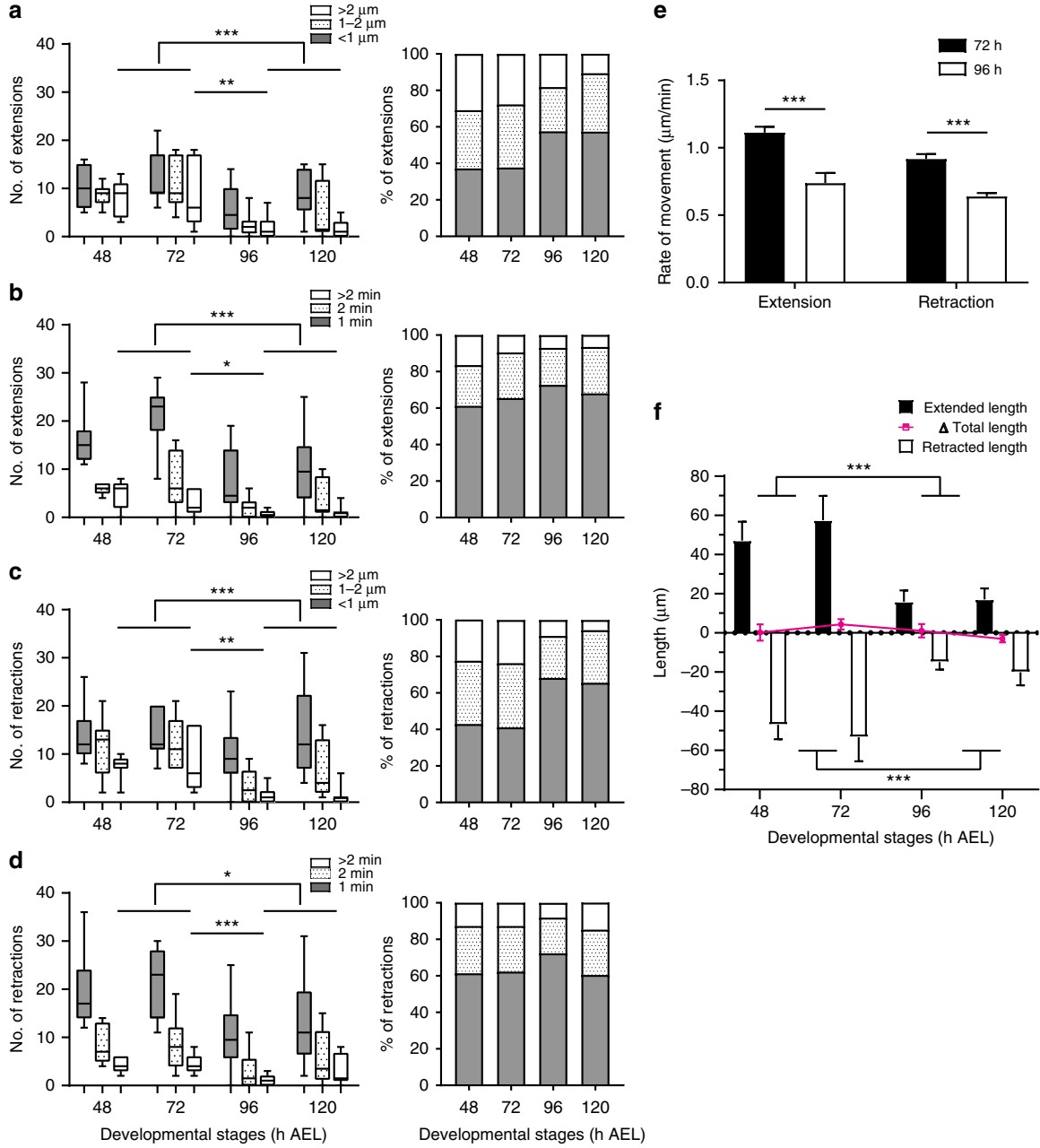

**Fig. 2** LNv dendrites change from dynamic to stable during development. **a**, **b** At all stages, most extensions are within 2 µm or lasted less than 2 min. More extension events that last longer than 2 min or travel further than 2 µm were observed in young neurons as compared to mature neurons. Distributions of distance and duration are shown. **c**, **d** Retractions behave similarly as extensions. Statistical significance was assessed by two-tailed Student's *t*-test for extensions/retractions with distance more than 2 µm or duration more than 2 min. **a** $P < 0.001$, 2nd vs 3rd instar; $P = 0.008$, 72 vs 96 h. **b** $P < 0.001$, 2nd vs 3rd instar; $P = 0.018$, 72 vs 96 h. **c** $P < 0.001$, 2nd vs 3rd instar; $P = 0.004$, 72 vs 96 h. **d** $P = 0.011$, 2nd vs 3rd instar; $P = 0.001$, 72 vs 96 h. **e** Dendrites in young LNvs (72 h) extend and retract faster than the ones in mature LNvs (96 h). Statistical significance was assessed by sum-of-squares *F* test of slopes generated by linear fitting of the distance vs. duration. For extension: slope is $1.113 \pm 0.044$ for 72 h, and $0.738 \pm 0.075$ for 96 h. For retraction: slope is $-0.917 \pm 0.037$ for 72 h, and $-0.639 \pm 0.025$ for 96 h; $P < 0.001$ for both comparisons. $n = 11$, 72 h; $n = 13$, 96 h. **f** Cumulatively, young neurons (48 and 72 h) extend and retract significantly more than mature neurons (96 and 120 h), without affecting the total dendritic length. Statistical significance was assessed by two-tailed Student's *t*-test. Extended length: $P < 0.001$; retracted length: $P < 0.001$. $n = 7, 7, 10$, and 8 for 48, 72, 96, and 120 h, respectively, for **a**–**d**, and **f**. Data are presented as box plot (box, 25–75%; center line, median) for **a**–**d**; Bar heights are means and error bars are SEM in **e**–**f**; ns, not significant; $*P < 0.05$, $**P < 0.01$, $***P < 0.001$

structural plasticity in regulating dendrite development and identify dynamic dendritic filopodia as its key cellular substrate.

**In vivo RNAi screen for regulators of dendrite dynamics.** Next, we sought to identify the molecular targets for experience-

dependent modification of dendrite dynamics using our newly established system. In late 3rd instar larvae (120 h AEL) cultured in LL conditions, almost all dynamic behaviors of LNvs dendrite branches are eliminated (Fig. 4b). This observation provides a strong phenotype for performing genetic screens for cell intrinsic factors involved in regulating dendrite dynamics.

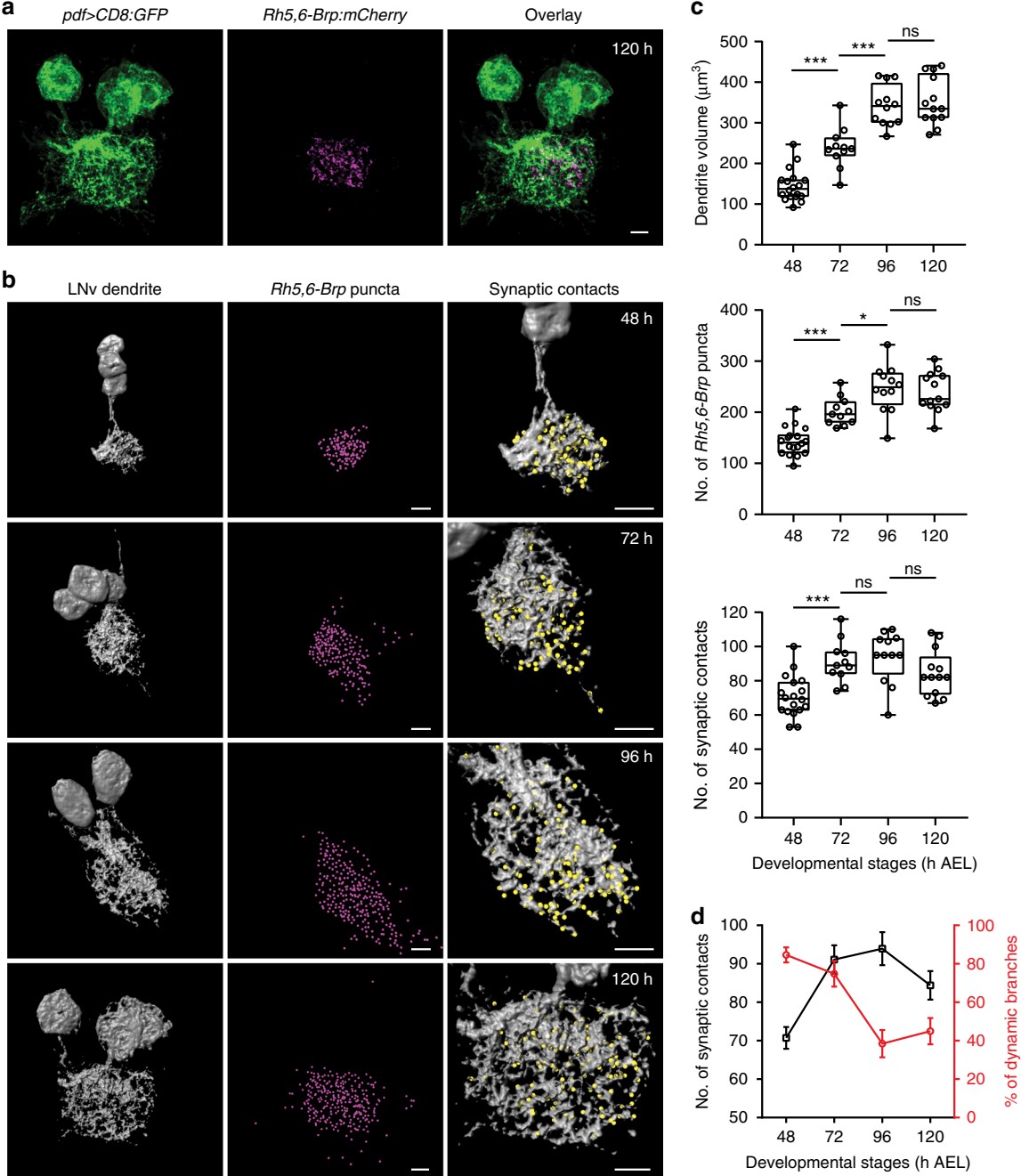

**Fig. 3** Temporal coupling of dendrite dynamics with synaptogenesis in LNvs. **a** Representative projected confocal images of the LNv dendrite (green) co-labeled with *Rh5,6-Brp:mCherry* (magenta). Scale bar, 5 μm. **b** The LNv dendrite volume (gray), the number of presynaptic terminals from the BN in the LON region (*Rh5,6-Brp* puncta, magenta spots) and the number of Brp puncta contacting the LNv dendrites (synaptic contacts, yellow spots) were analyzed using 3D reconstructions. Representative images for samples collected at the indicated time points are shown. Scale bar, 5 μm. **c** LNv dendrite volume (Top) increases during development until 96 h; *Rh5,6-Brp* puncta (Middle) shows similar trend and plateaus at 96 h; The number of synaptic contacts (Bottom) increases significantly from 48 to 72 h, then remains steady. $n = 18, 11, 12$, and 13 for 48, 72, 96, and 120 h, respectively. Statistical significance was assessed by one-way ANOVA followed by Tukey's post hoc test. Volume (Top), ANOVA: $F_{3, 50} = 60.19$, $P < 0.001$; Tukey's: $P < 0.001$, 48 vs 72 h and 72 vs 96 h; $P = 0.949$, 96 vs 120 h. *Rh5,6-Brp* puncta (Middle), ANOVA: $F_{3, 50} = 30.03$, $P < 0.001$; Tukey's: $P < 0.001$, 48 vs 72 h; $P = 0.015$, 72 vs 96 h; $P = 0.963$, 96 vs 120 h. Synaptic contacts (Bottom), ANOVA: $F_{3, 50} = 9.465$, $P < 0.001$; Tukey's: $P = 0.001$, 48 vs 72 h; $P = 0.955$, 72 vs 96 h; $P = 0.278$, 96 vs 120 h. Data are presented as box plot (box, 25–75%; center line, median) overlaid with dot plot (individual data points). ns, not significant; *$P < 0.05$, ***$P < 0.001$. **d** The number of synaptic contacts (black) reaches saturation at 72 h, coinciding with the peak of LNv dendrite dynamics (red). Trend lines were generated using data from Figs. 1c, 3c. Error bars represent SEMs

We performed LNv-specific in vivo transgenic RNAi screens on a collection of RNAi lines targeting 134 genes, including cytoskeleton components and motor proteins, as well as signaling molecules related to cytoskeleton remodeling and synapse formation (Supplementary Data 1). Larvae co-expressing membrane-targeted GFP, a transgenic RNAi construct and Dicer2 driven by Pdf-Gal4, were cultured under LD or LL conditions. From the same genotype, we characterized dendrite volume

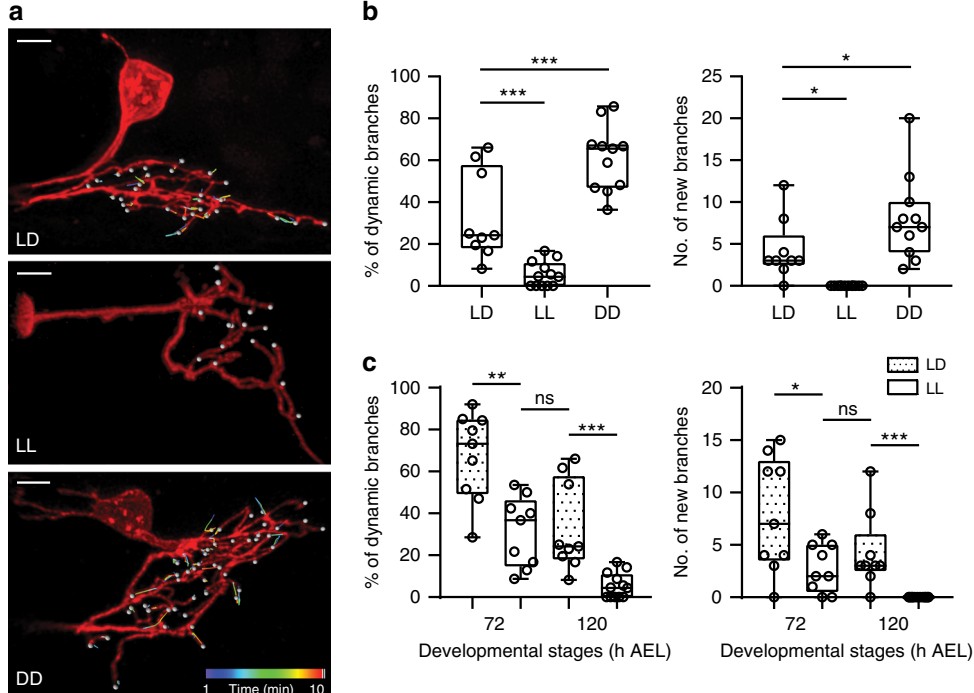

**Fig. 4** Experience-dependent modification of the dynamic state of LNv dendrites. **a** Representative tracking trajectories generated by movements of dynamic filopodia in single-labeled LNvs collected at 120 h AEL from larvae cultured in light dark (LD), constant light (LL), or constant darkness (DD). Scale bar, 5 μm. **b** Differential visual experiences provided by LL or DD during larval development generate opposite effects on dendrite dynamics in LNvs. Compared to LD, the percentage of dynamic branches, as well as the number of newly appeared branches, significantly increases in DD but decreased in LL. $n = 9$, 12, and 11 for LD, LL, and DD, respectively. Statistical significance was assessed by one-way ANOVA with Dunnett post hoc test. Dynamic branches (Left), ANOVA: $F_{2, 29} = 39.441$, $P < 0.001$; Dunnett: $P < 0.001$, LD vs others. New branches (Right), ANOVA: $F_{2, 29} = 14.85$, $P < 0.001$; Dunnett: $P = 0.021$, LD vs LL; $P = 0.044$, LD vs DD. **c** Compared to the LD group, young LNvs (72 h) cultured in LL have significantly reduced dendrite dynamics, close to the levels observed in mature LNv (120 h) cultured in LD. $n = 9$, 9, 9, and 12 for 72LD, 72LL, 120LD, and 120LL, respectively. Statistical significance was assessed by two-tailed Student's t-test. Dynamic branches (Left): $t_{16} = 4.012$, $P = 0.001$, 72LD vs 72LL; $t_{19} = 4.297$, $P < 0.001$, 120LD vs 120LL; $t_{16} = 0.194$, $P = 0.849$, 72LL vs 120LD. New branches (Right): $t_{16} = 2.59$, $P = 0.02$, 72LD vs 72LL; $t_{19} = 4.101$, $P < 0.001$, 120LD vs 120LL; $t_{16} = 1.017$, $P = 0.324$, 72LL vs 120LD. Data are presented as box plot (box, 25–75%; center line, median) overlaid with dot plot (individual data points). ns, not significant; *$P < 0.05$, **$P < 0.01$, ***$P < 0.001$

through confocal imaging on fixed tissues and performed dendrite dynamic studies using time-lapse live imaging. Using this approach, we identified several candidate proteins, including Cdc42 and Rho1, small GTPases that regulate actin dynamics, as well as Fas2, a synaptic cell adhesion molecule, and Amph, a BAR domain-containing protein.

Knocking-down Cdc42 and Rho1 leads to elevated dendrite dynamics accompanied by reductions in LNv dendritic arbor size, replicating previous findings in vertebrate neurons, in which Cdc42 and Rho1 have critical functions in regulating the motility and prevalence of dendrite filopodia[4,17,33,34]. Similar phenotypes are observed by expressing RNAi targeting Fas2. To validate the function of Fas2 in regulating LNv dendrite morphogenesis, we tested a transgene encoding a dominant-negative form of Fas2 (Fas2Δ3), which has a c-terminal deletion that abolishes its interaction with Dlg and alters its synaptic localization[35,36]. Expressing Fas2Δ3 in LNvs elevates the branch dynamics while reducing the dendrite volume and the number of synaptic contacts between the BN and LNvs (Supplementary Figure 6), consistent with the known function of Fas2 in maintaining synaptic organization and stability[37]. Although knocking-down either Cdc42, Rho1, or Fas2 alters LNv dendrite dynamics, it also appears to impact multiple aspects of neuronal morphology. In addition, Fas2 and Rho1 activity are known to be involved in the circadian regulation of axon morphology in the small LNvs (s-LNvs) of adult flies[38,39].

In search of a candidate that produces a dynamic-specific phenotype, we found that knocking-down *amphiphysin* (*amph*) generates numerous dynamic events at 120 h AEL in LL conditions, while the morphology and size of LNv dendrites remain largely unaffected (Fig. 6a, b, Supplementary Figure 7). The separation of the dynamic phenotype from the deficit in dendrite growth supports our model in which dynamic filopodia are tightly associated with synaptogenesis, but do not directly contribute to changes in total dendrite length. Furthermore, this finding provides us a molecular handle to study the differential regulation of dendrite dynamics and growth.

**Amph regulates LNv dendrite dynamics cell-autonomously.** Amph was not previously characterized in the *Drosophila* central synapse. To understand its function in regulating dendrite development, we first validated RNAi screen results on Amph using single-labeled neurons. In LD conditions, knocking-down Amph in LNvs also leads to increases in the percentage of dynamic branches and the number of newly appeared branches, while there is no effect on either dendrite volume or the number of synaptic contacts between LNvs and the BN (Fig. 6c–e, Supplementary Movies 9 and 10). These results suggest that Amph is a part of the regulatory mechanisms that target the prevalence of dynamic filopodia. Notably, elevated dendrite dynamics generated by the reduction of Amph did not promote synaptogenesis or dendrite growth. Therefore, although exploratory dendritic

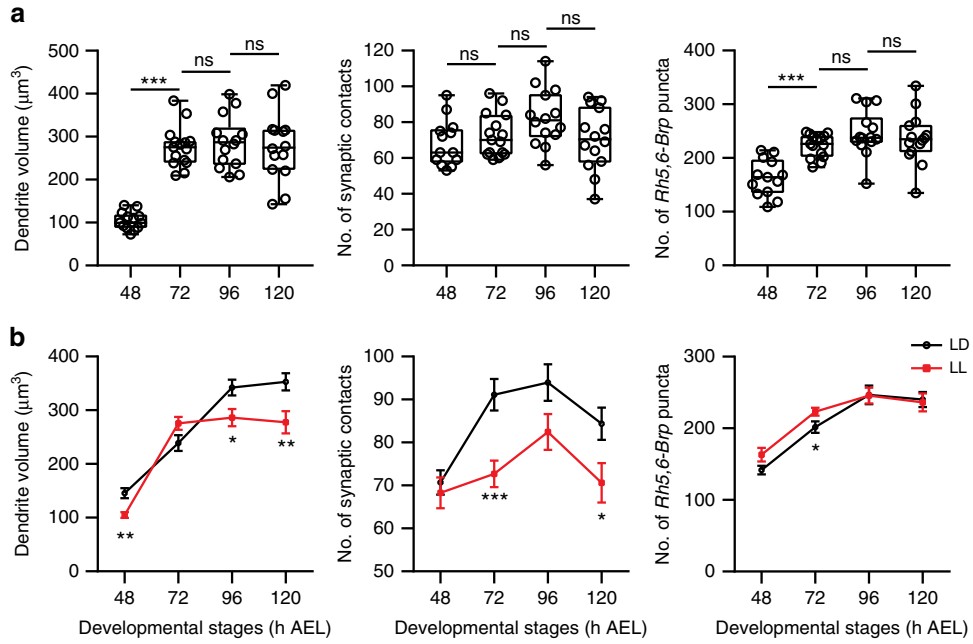

**Fig. 5** Experience-dependent modifications of dendrite growth and synaptogenesis in LNvs. **a** LL conditions alter the temporal profile of dendrite growth and synapse formation. LNv dendrite volume and the number of presynaptic terminals from the BN in the LON region (Rh5,6-Brp puncta) plateaued at 72 h, while the number of Brp puncta contacting the LNv dendrites (synaptic contacts) remains unchanged from 48 h. $n = 13, 15, 14, 14$ for 48, 72, 96, 120 h, respectively. Statistical significance was assessed by one-way ANOVA with Tukey's post hoc test. Volume (Left), ANOVA: $F_{3, 52} = 32.38$, $P < 0.001$; Tukey's: $P < 0.001$, 48 vs 72 h; $P = 0.956$, 72 vs 96 h; $P = 0.974$, 96 vs 120 h. Synaptic contacts (Middle), ANOVA: $F_{3, 52} = 2.493$, $P = 0.07$; Tukey's: $P = 0.859$, 48 vs 72 h; $P = 0.284$, 72 vs 96 h; $P = 0.155$, 96 vs 120 h. Rh5,6-Brp puncta (Right), ANOVA: $F_{3, 52} = 12.96$, $P < 0.001$; Tukey's: $P < 0.001$, 48 vs 72 h; $P = 0.369$, 72 vs 96 h; $P = 0.906$, 96 vs 120 h. Data are presented as box plot (box, 25–75%; center line, median) overlaid with dot plot (individual data points). **b** Compared to the LD condition, LL produces a premature termination of synapse formation and dendrite volume expansion, without affecting the number of presynaptic terminals from the BN. Trend lines of the three parameters from LD or LL were plotted. Error bars represent SEMs. LD and LL compared at each developmental stage by two-tailed Student's t-test. Volume (Left), 48 h: $t_{29} = 3.398$, $P = 0.002$; 96 h: $t_{24} = 2.544$, $P = 0.018$; 120 h: $t_{25} = 2.801$, $P = 0.01$. Synaptic contacts (Middle), 72 h: $t_{24} = 3.825$, $P = 0.001$; 120 h: $t_{25} = 2.298$, $P = 0.03$. Rh5,6-Brp puncta (Right), 72 h: $t_{24} = 2.253$, $P = 0.034$. ns, not significant; *$P < 0.05$, **$P < 0.01$, ***$P < 0.001$

filopodia support synaptogenesis, the upregulation of dendrite dynamics in postsynaptic neurons is not sufficient to increase the synapse formation.

Next, we performed phenotypic analysis on *amph* loss-of-function mutants that contain deletions eliminating all isoforms (Supplementary Figure 8a)[26,28,40]. We tested two null alleles, *amph⁵ᴱ³* and *amph²⁶*, and observed in both genotypes a significant increase in the number of newly appeared branches at 120 h AEL in LL conditions (Fig. 7c. Supplementary Figure 8c), similar to the RNAi screen results. However, although mutant larvae develop at a regular pace and have normal-sized brains, Amph null mutants (*amph⁵ᴱ³*, *Amph⁻/⁻*) show severe reductions in the volume of LNv dendrites and the number of BN presynaptic terminals in the LON as well as in the number of synaptic contacts between LNvs and the BN (Fig. 7a, b), which were not observed in the LNv-specific Amph RNAi experiments. These strong mutant phenotypes reveal previously unidentified functions of Amph in synaptogenesis and dendrite development.

To determine if the deficits observed in *amph* null mutants reflect cell-autonomous requirements of Amph in LNvs, we performed rescue experiments by introducing an Amph transgene driven by LNv-specific Pdf-Gal4 into the mutant genetic background[40]. The excessive dynamic phenotype of the *amph* mutant is partially rescued by this approach, while the number of synaptic contacts and dendrite volume remain at the same low level as the mutant (Fig. 7b, c). It is possible that the level and timing of the transgene expression are not sufficient to rescue the mutant's deficits in synaptogenesis and dendrite development.

However, based on the broad distribution of Amph in the larval CNS and the severe reduction in the number of total BN presynaptic terminals in the LON[25,28], this is more likely due to the non-autonomous functions of Amph that occur early in larval development. Nevertheless, both LNv-specific Amph RNAi knockdown and the Amph transgene rescue experiments support the cell-autonomous function of Amph in regulating dendrite dynamics.

To characterize the expression pattern of the Amph protein, we obtained an anti-Amph antibody for immunostaining and an Amph enhancer Gal4 line to drive CD8:GFP expression[41]. Results generated by both approaches confirmed the wide distributions of Amph in larval CNS as previously reported and its expression in LNvs[28]. With thin optical sections, we observed Amph staining on LNv dendrites, where there was no clear overlap of Amph signal with Rh5,6-Brp:mCherry puncta, suggesting that Amph is not enriched in the presynaptic terminals of the BN (Supplementary Figure 10). Furthermore, using quantitative real-time PCR analysis, we found that the level of *amph* transcript was significantly increased at the late stages of larval development (Supplementary Figure 8b), supporting its potential function in promoting dendrite stabilization in mature neurons. However, overexpressing a full length Amph transgene driven by Pdf-Gal4 did not affect either the dynamics or the morphology of the LNv dendrite (Supplementary Figure 9a, b)[28]. Although reducing Amph leads to elevated dendrite dynamics, increasing its expression alone may not be sufficient to promote stabilization of the dendritic arbors. It is also possible that the Pdf-Gal4 driven

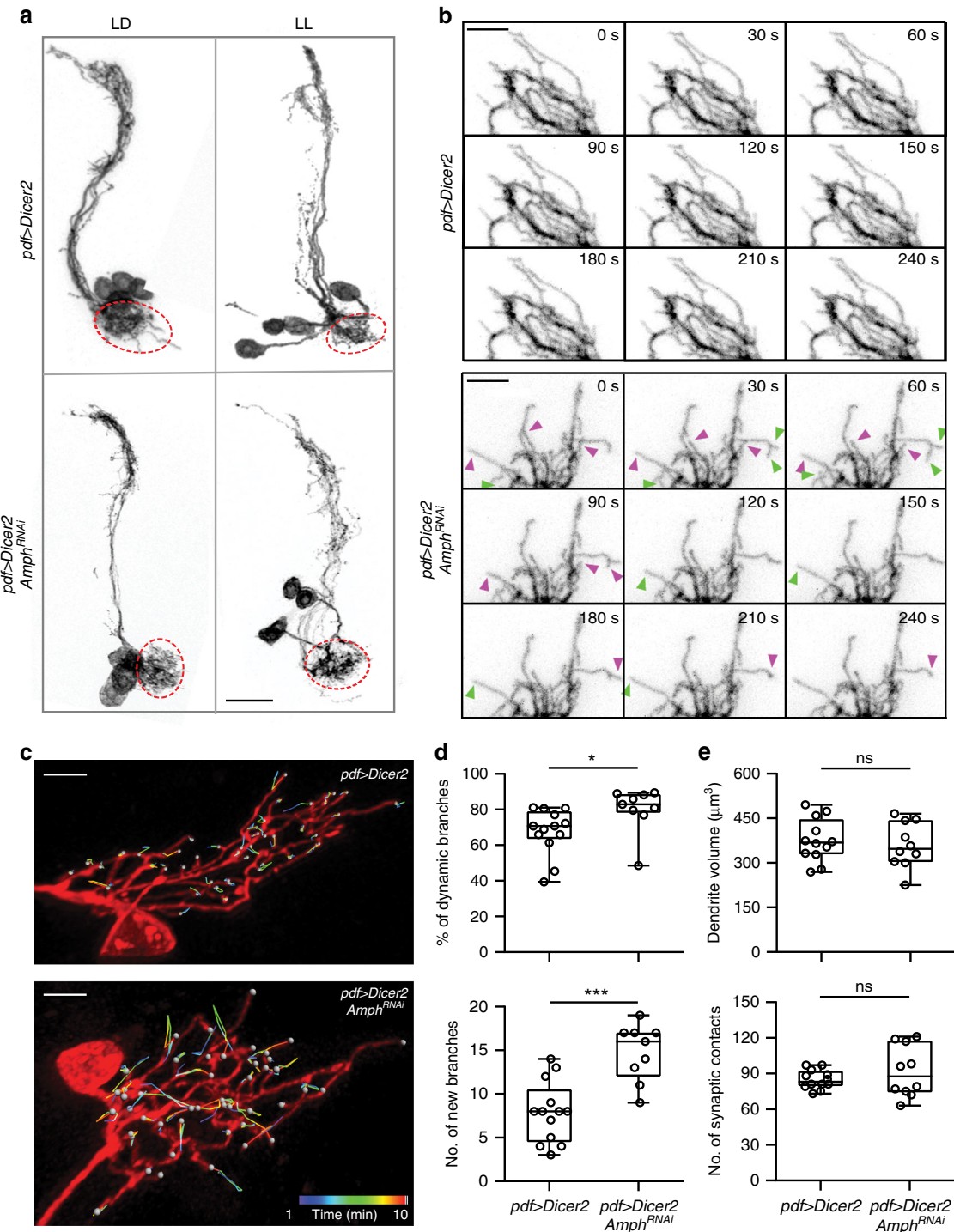

**Fig. 6** RNAi knockdown of Amph elevates dendritic dynamics in LNv without affecting dendrite morphology. **a** Knocking-down Amph expression does not affect LNv dendrite morphology in both LD and LL conditions. Representative projected confocal images of LNvs collected at 120 h AEL are shown. Genotypes and culture conditions were as indicated. Dendritic field are circled with dashed lines (red). **b** Knocking-down Amph led to increased number of dynamic events at 120 h in LL. Representative cropped frames from live imaging series are shown. Extensions (green) and retractions (magenta) were only observed in the LNv expressing RNAi targeting Amph. Scale bar, 5 μm. **c** Knocking-down Amph in LNvs increases dendrite dynamics without affecting dendrite morphology. Representative tracking trajectories generated by movements of dynamic filopodia in single-labeled LNvs are shown. Genotypes are as indicated. **d** Compared to the control, both the percentage of dynamic branches and the number of newly appeared branches increase significantly in the Amph knockdown group. $n = 13$, $pdf > Dicer2$; $n = 9$, $pdf > Dicer2$, $Amph^{RNAi}$. Statistical significance was assessed by two-tailed Student's $t$-test. Dynamic branches (Top): $t_{20} = 2.243$, $P = 0.036$. New branches (Bottom): $t_{20} = 4.657$, $P < 0.001$. **e** Knocking-down Amph does not affect LNv dendrite volume or the number of Brp puncta contacting LNv dendrites (synaptic contacts). $n = 12$, $pdf > Dicer2$; $n = 10$, $pdf > Dicer2$, $Amph^{RNAi}$. Statistical significance was assessed by two-tailed Student's $t$-test. Volume (Top): $t_{20} = 0.503$, $P = 0.62$. Synaptic contacts (Bottom): $t_{20} = 1.015$, $P = 0.322$. Data are presented as box plot (box, 25–75%; center line, median) overlaid with dot plot (individual data points). ns, not significant; *$P < 0.05$, **$P < 0.01$, ***$P < 0.001$

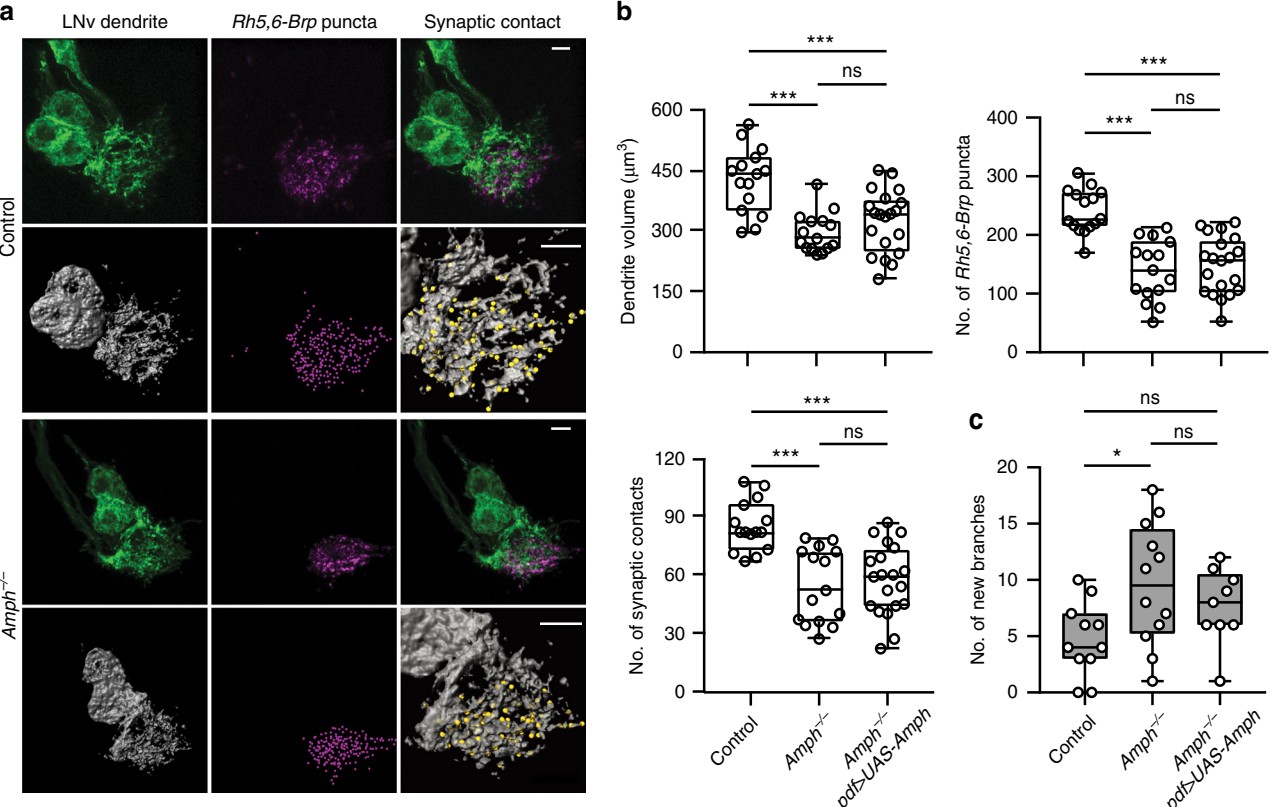

**Fig. 7** *Drosophila* Amph regulates the dynamic state of LNv dendrites cell-autonomously. **a** Compared to the background control, *amph* null mutants (*Amph*$^{-/-}$) have significant reductions of dendrite volume (green or gray), the number of presynaptic terminals (magenta), and synaptic contacts (yellow spots). Representative projected confocal images (Top) or 3D reconstructions (Bottom) are shown. Genotypes are as indicated. Scale bar, 5 μm. **b** The reduction of dendrite volume, *Rh5,6-Brp* puncta or synaptic contacts, in *Amph*$^{-/-}$ was not rescued by introducing an Amph transgene into LNvs (*Amph*$^{-/-}$, *pdf > UAS-Amph*). $n$ = 15, 15, and 20 for Control, *Amph*$^{-/-}$, and *Amph*$^{-/-}$, *Pdf > UAS-Amph* respectively. Statistical significance was assessed by one-way ANOVA with Tukey's post hoc test. Volume (Top left), ANOVA: $F_{2, 47}$ = 14.21, $P < 0.001$; Tukey's: $P < 0.001$, Control vs others; $P = 0.468$, *Amph*$^{-/-}$ vs *Amph*$^{-/-}$, *Pdf > UAS-Amph*. *Rh5,6-Brp* puncta (Top right), ANOVA: $F_{2, 47}$ = 22.26, $P < 0.001$; Tukey's: $P < 0.001$, Control vs others; $P = 0.842$, *Amph*$^{-/-}$ vs *Amph*$^{-/-}$, *Pdf > UAS-Amph*. Synaptic contacts (Bottom left), ANOVA: $F_{2, 47}$ = 14.99, $P < 0.001$; Tukey's: $P < 0.001$, Control vs others; $P = 0.876$, *Amph*$^{-/-}$ vs *Amph*$^{-/-}$, *Pdf > UAS-Amph*. **c** Elevated dendrite dynamics in *Amph*$^{-/-}$ is partially rescued by reintroducing Amph into LNvs. $n$ = 11, 12, and 9 for Control, *Amph*$^{-/-}$, and *Amph*$^{-/-}$, *Pdf > UAS-Amph* respectively. Quantifications were from all four LNvs of larvae raised in LL to 120 h AEL. Statistical significance was assessed by one-way ANOVA with Tukey's post hoc test. ANOVA: $F_{2, 29}$ = 3.82, $P = 0.034$; Tukey's: $P = 0.026$, Control vs *Amph*$^{-/-}$; $P = 0.284$, Control vs *Amph*$^{-/-}$, *Pdf > UAS-Amph*; $P = 0.565$, *Amph*$^{-/-}$ vs *Amph*$^{-/-}$, *Pdf > UAS-Amph*. Scale bar, 5 μm. Data are presented as box plot (box, 25–75%; center line, median) overlaid with dot plot (individual data points). ns, not significant; *$P < 0.05$, **$P < 0.01$, ***$P < 0.001$

expression of the Amph transgene did not achieve either the necessary protein level or the proper localization required for the dendrite stabilization phenotype.

**Amph regulates the postsynaptic organization and function**. How does Amph regulate dendrite dynamics without changing the number of synapses? To address this question, we tested whether Amph is involved in regulating synaptic functions. The LON is not accessible by electrophysiological measurements. Therefore, we analyzed the synaptic functions in LNvs through optic recordings using a genetically-encoded calcium sensor, GCaMP6[23]. LNvs respond to light-induced synaptic input from the BN and produce robust calcium signals at the axonal terminal region (Fig. 8a). Using this calcium imaging approach, we detected significantly reduced physiological responses in LNvs expressing Amph RNAi as compared to the control group, suggesting that the synaptic transmission between the BN to LNvs is impaired when the Amph level is reduced (Fig. 8b, c).

Previous studies in the *Drosophila* NMJ have shown that Amph influences the localization of the postsynaptic proteins Dlg, Lgl, and Scrib. Amph is also involved in the membrane integration of

the synaptic adhesion molecule Fas2[25,37]. Therefore, it is conceivable that Amph acts together with these postsynaptic components in regulating the organization and function of the central synapse. Since Fas2 is another candidate identified in our genetic screen, we went on to test whether the dendritic distribution of Fas2 is affected by reducing Amph.

We examined the membrane integration of Fas2 on LNv dendrites using a transgene expressing a YFP fusion protein containing the extracellular domain of Fas2 (extra-Fas2:YFP)[42]. This genetic tool allowed us to monitor the mobility and the membrane localization of Fas2 without affecting LNv dendrite morphology (Supplementary Figure 11). In LNvs, the extra-Fas2: YFP signals were concentrated in the synaptic regions, including axon terminals and dendritic arbors, but not in the plasma membrane or the axonal tract, which contain no synapses. In contrast, another YFP fusion protein constructed using the intracellular domain of Fas2 (intra-Fas2:YFP) shows a diffused pattern of expression and can be found throughout the membrane structures of LNvs (Supplementary Figure 11). We then reconstructed the distribution of extra-Fas2:YFP on LNv dendritic arbors in 3D. In contrast to the membrane-bound CD8:

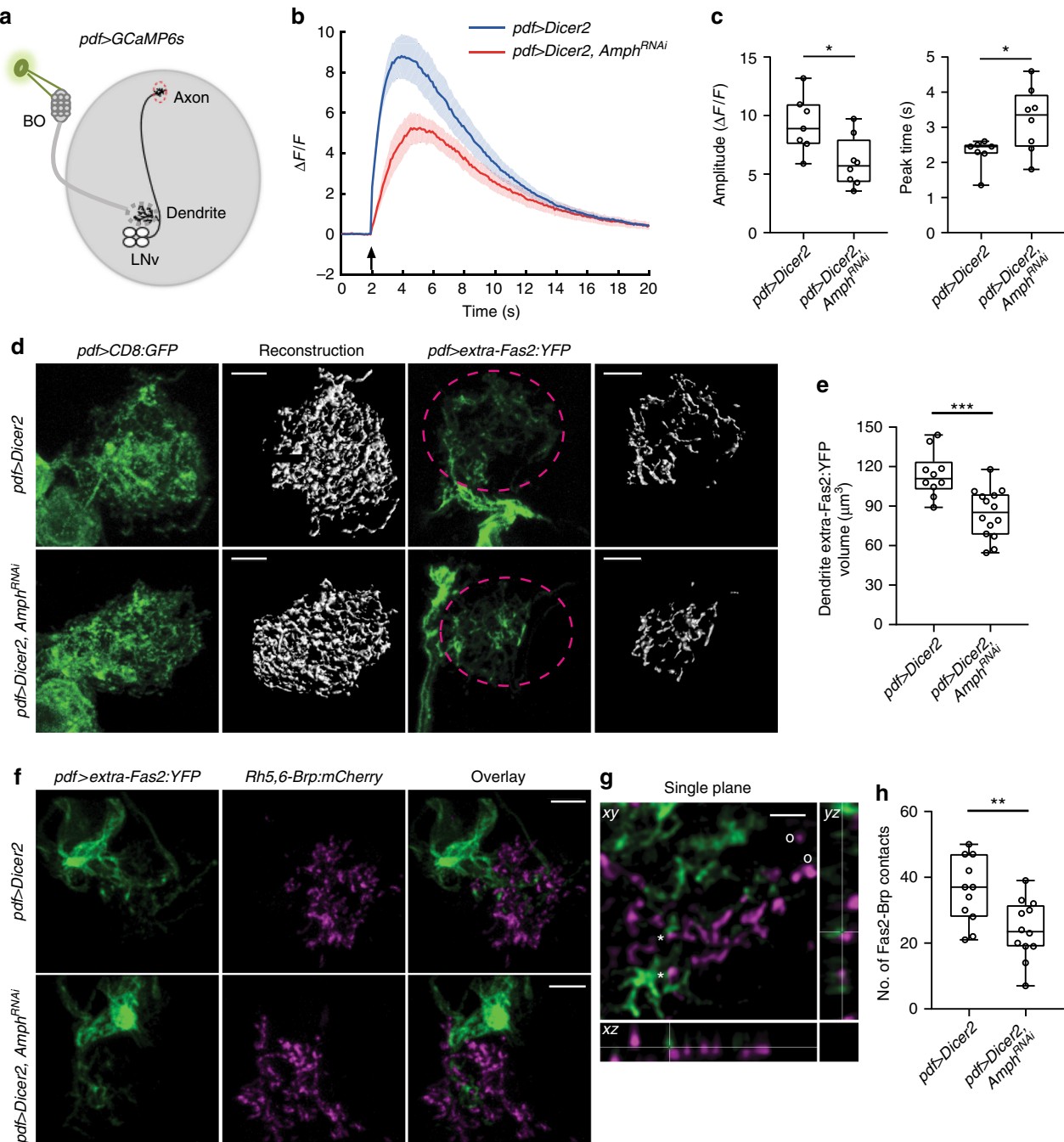

**Fig. 8** Amph is required for postsynaptic organization and normal synaptic transmission. **a** Schematic diagram for calcium imaging experiments. BO-mediated synaptic transmission is received by LNv dendrites (dashed gray circle) and measured by the changes of GCaMP6s signals in the axonal terminal region of LNvs (dashed red circle). **b** Knocking-down Amph led to reductions in light-induced responses in LNvs. The changes in GCaMP signal (ΔF/F) were plotted. Data are presented as line graph showing Mean ± SEM. Arrow indicates the delivery of the light stimulation. **c** Quantification reveals reduced amplitude and delayed peak time of calcium responses in LNvs expressing Amph RNAi. $n = 7$, $pdf > Dicer2$; $n = 8$, $pdf > Dicer2$, $Amph^{RNAi}$. Statistical significance was assessed by two-tailed Student's $t$-test. Amplitude: $t_{13} = 2.742$, $P = 0.017$. Peak time: $t_{13} = 2.491$, $P = 0.027$. **d** Disrupted dendritic Fas2 distribution in LNvs expressing Amph RNAi. While CD8:GFP ($pdf > CD8:GFP$, left) shows no change, the volume of dendritic extra-Fas2:YFP ($pdf > extra-Fas2:YFP$, right) is reduced in LNvs with Amph knockdown. Representative projected confocal images (green) and 3D reconstructions (gray) are shown. Scale bar, 5 μm. **e** Quantification of dendritic extra-Fas2:YFP based on 3D reconstructions. $n = 10$, $pdf > Dicer2$; $n = 14$, $pdf > Dicer2$, $Amph^{RNAi}$. Statistical significance was assessed by two-tailed unpaired Student's $t$-test. $t_{22} = 3.969$, $P < 0.001$. **f–h** Amph knockdown reduces the synaptic distribution of Fas2. **f** Representative projected confocal images of LNvs are shown. Scale bar, 5 μm. Genotypes are as indicated. **g** A representative single optic section image illustrating the dendritic Fas2 (extra-Fas2:YFP, green) contacting the presynaptic site marked by Brp ($Rh5,6-Brp$, magenta). Contacts are indicated by *, non-contacts by o. Scale bar, 2 μm. **h** Quantifications of the number of contacts between Brp puncta and extra-Fas2:YFP. Knocking-down Amph led to reduced number of Fas2-Brp contacts. $n = 11$, $pdf > Dicer2$; $n = 12$, $pdf > Dicer2$, $Amph^{RNAi}$. Statistical significance was assessed by two-tailed Student's $t$-test. $t_{21} = 2.988$, $P = 0.007$. Data are presented as box plot (box, 25–75%; center line, median) overlaid with dot plot (individual data points). ns, not significant; *$P < 0.05$, **$P < 0.01$, ***$P < 0.001$

GFP, which were evenly localized on dendrites in both control and Amph knockdown LNvs, the extra-Fas2:YFP occupancy on the dendritic arbors, as reflected by the volume of the reconstructed YFP signals, decreases significantly in the Amph knockdown group (Fig. 8d, e).

To confirm extra-Fas2:YFP's synaptic distribution, we examined its localization against the presynaptic Rh5,6-Brp:mCherry puncta and observed close associations of YFP and mCherry signals throughout the LNv dendrite labeled by extra-Fas2:YFP (Fig. 8f–h). In addition, in LNvs with an Amph knockdown, the number of Fas2-Brp contacts was significantly reduced compared to controls (Fig. 8f–h), supporting the role for Amph in regulating the dendritic distribution of Fas2. With a well-established role in synapse maintenance and plasticity, Fas2's distribution impacts the stability of synapses as well as activity-induced structural and functional plasticity[37,43], which potentially contribute to the phenotypes we observed in Amph knockdown LNvs.

Taken together, our cell-specific RNAi analyses validate the connections between Amph and Fas2 in the postsynaptic compartment of CNS neurons. Furthermore, our results demonstrate that reducing Amph in mature neurons dampens their physiological responses and decreases the dendritic distribution of synaptic cell adhesion molecules, suggesting that the elevated dendrite dynamics maybe caused by impaired synaptic organization. These observations further indicate Amph as an important postsynaptic component that is required for regulating dendrite dynamics and dendrite maturation during development and suggest its role in maintaining structure and function stability of the synapse in mature neurons.

## Discussion

Developing neural circuits exhibit remarkable capacities for structural and functional plasticity[18,44]. How these capacities are determined by genetic programs and experience and how they are regulated during development are key questions we are addressing by genetic analyses in a simple fly sensory circuit. In this study, we demonstrate experience-dependent regulation of dendrite dynamics and provide evidence for sensory experience globally regulating the pace of dendrite maturation and modifying dendrite's capacity for synaptogenesis and growth during early development. Using our model system, we performed systematic analyses of dynamic dendritic filopodia, dendrite growth, and synapse formation throughout larval development and made the following observations:

First, there is a large degree of similarity between the dendritic filopodia of *Drosophila* larval LNvs and those in developing vertebrate neurons. The properties of LNv dendritic filopodia closely resemble the ones found in aspiny chick RGCs, with similar rates of movement, around 1 µm/min, and a short range of action, mainly around 1–2 µm, suggesting that evolutionarily conserved cellular machinery controls the behavior of dendritic filopodia[45]. Second, the composition of the population of branches on LNv dendritic arbors is modified during development, with most dynamic branches supporting synaptogenesis in young LNvs, followed by a transition into the mature state when more branches are stable. While local calcium transients can induce stabilization of individual filopodia or branches[9,46], our results reveal that intrinsic and global mechanisms also play important roles in defining the dynamic state of the dendritic arbor. Third, the dynamic state of the dendrite does not influence its growth directly. Our studies provide clarifications on the connection between dendrite dynamics and growth. Both our live imaging analyses and genetic studies suggest that these two processes are differentially regulated and the dynamic state of dendrites indirectly impacts growth by regulating the capacity for synapse

formation. Importantly, despite the opposite changes in dendrite dynamics, LNvs collected from constant light conditions and Amph null mutants both show reductions in synapse number and dendrite volume. These observations fit the predictions of the synaptotrophic hypothesis, which suggests that synaptic input is the main drive that controls elaborations of dendritic arbors and that the lack of synapses would hinder dendrite growth[47,48]. Lastly, we found that developmental control of the dynamic state is critical for the stability and function of the circuit. Elevated dendrite dynamics in mature neurons are not sufficient to promote synapse formation and are signs of weakened synaptic connection and function.

Our genetic studies focused on the role of Amph as a regulator for dendrite dynamics and a component of the postsynaptic compartment in the fly CNS. Due to its broad distribution and versatile roles in organizing membranous structures, Amph is potentially involved in multiple aspects of neuronal structure and functions through both autonomous and non-autonomous actions. This is clearly demonstrated in this study by different phenotypes generated by the loss-of-function Amph mutants and the LNv-specific Amph knockdown approach. Notably, Amph knockdown produced a specific dendrite dynamic phenotype without changes in synapse number and dendrite volume. In addition, this approach also helped us identifying the function of Amph in regulating the dendritic distribution of Fas2 and maintaining the structural and functional stability of the synapse. These findings provide molecular insights for Amph's neuronal functions. Amph has homologs from yeast to human with the conserved BAR domain functions as a protein dimerization, curvature-sensing, and membrane-binding module[24]. In addition to Amph I, which regulates vesicle recycling and endocytosis at the presynaptic terminals in mammalian brains, multiple isoforms of vertebrate Amph with distinct tissue distributions and diverse functions have been identified[27]. Several lines of evidence suggest that *Drosophila* Amph is an ortholog of vertebrate Amph II[25,40]. Interestingly, human Amph II, also known as bridging integrator 1 (BIN1), has recently been identified as the second most important risk locus for late onset Alzheimer's disease (AD) by genome wide association studies[49]. Our studies establish a link between reduced Amph level in mature neurons with elevated dendrite dynamics and impaired synaptic function, potentially providing a postsynaptic mechanism for the dysfunction and loss of synapses featured in the early stage of AD[50].

Sensory experience has a strong influence on the dynamic state of the dendrite, with the chronic elevation of light inputs promoting the maturation of LNv dendrites and suppressing the capacity for filopodia formation. These changes, in turn, lead to a reduction in synapse number and dendrite expansion. This observation provides a plausible cellular mechanism for the activity-dependent regulation of dendrite arbor size that we and others have observed[23,51] and supports structural homeostasis as an important component of developmental plasticity. Homeostatic structural plasticity has a long-lasting impact on neuronal intrinsic excitability and circuit properties, but remains largely uncharacterized[52]. Our studies reveal experience-dependent regulation of dendrite dynamics and its strong impact on synaptogenesis and dendrite development. These findings provide a basis for future molecular studies on developmental and activity-dependent regulation of wiring plasticity through homeostatic mechanisms.

## Methods

**Fly stocks.** Mutant and transgenic lines used are as follows: (1) *pdf-Gal4; UAS-CD8:GFP*, (2) *hs-flp; pdf-Gal4; UAS-FRT-CD2-stop-FRT-CD8:GFP*, (3) *pdf-LexA, LexAop-CD8:tdTomato*, (4) *Rh5,6-Brp:mCherry (Rh5-Brp:mCherry, Rh6-Brp:mCherry)*, (5) *UAS-Dicer2* (stock number 24651), (6) *UAS-Amph^RNAi* (stock

number 53971), (7) UAS-Amph, (8) amph[5E3], (9) amph[26] (stock number 6498), (10) Amph[EP2175] (stock number 17236), (11) UAS-extra-Fas2:YFP (stock number 116988), (12) UAS-intra-Fas2:YFP (stock number 116989), (13) UAS-GCaMP6s (stock number 42746), (14) UAS-Fas2Δ3 (stock number 36285), (15) Amph[MI08903-TG4] (stock number 77794). Stock 1, 2, 3 were as described[23]. Stock 4 was from Chi-Hon Lee. Stock 5, 6, 9, 10, 13, 14, and 15 were from the Bloomington Stock Center. Stock 11 and 12 were from Kyoto Stock Center. Stock 7 and 8 were from Dr. Gabrielle Boulianne. For a full list of fly genotypes by figures, see Supplementary Methods.

**Fly rearing and heat shock.** Fly stocks are maintained in the standard medium in circadian and humidity controlled 25 °C incubators with a standard 12 h light:12 h dark schedule. Light intensity in the incubator is around ~1000 lux. Collections of developmental staged larvae and LNv single labeling were as described[23]. Briefly, embryos with desired genotypes were collected at 2 h intervals on grape juice plates with yeast supplement. To get single-labeled LNv, hs-flp; pdf-Gal4; UAS-FRT-CD2-stop-FRT-CD8:GFP flies were crossed to various transgenes and embryos were collected at 2 h intervals. Newly hatched larvae (~24 h AEL) were then heat shocked at 37 °C for two 40 min sessions with a 40 min recovery period in between.

**qRT-PCR analysis.** Larval brains (n = 15 for each condition) was dissected in PBS at indicated developmental stages. Total RNA was isolated by a Quick-RNA MiniPrep kit (R1054, Zymo Research). Reverse transcription was performed with oligo dT-primers using SuperScript® III First-Strand Synthesis System (18080051, Invitrogen) and Quantitative Real-Time PCR (qRT-PCR) using the cDNA was performed with SsoAdvanced Universal SYBR® Green Supermix (1725271, Bio-Rad Laboratories) on a Bio-Rad CFX96 Real-Time PCR machine. Experiment was independently repeated four times. All samples and standards were assayed in triplicates. Expression levels of Amph were normalized to those of GAPDH.
Primers for qRT-PCR:
Amph: ACTAAGACCAGCACAGGCAC, TCACCCGGTACAGAACTCCA;
GAPDH: TAAATTCGACTCGACTCACGGT, CTCCACCACATACTCGGCTC.

**Confocal and two-photon imaging.** The procedures for dissection, fixation, and immunochemistry on larval brains were as follows. Larval brains were dissected in PBS and fixed in 4% PFA/PBS at room temperature for 30 min, followed by washing in PBST (0.3% Triton-X 100 in PBS) and incubating in the primary antibody overnight at 4 °C. On the next day, the brains were washed with PBST and incubated in the secondary antibody at room temperature for 1 h before final washes in PBST and mounting on the slide with the antifade mounting solution (SlowFade Antifade kit, Life Technologies S2828). Primary antibodies used was rabbit anti-Amph (a gift from Dr. Gabrielle Boulianne, 1:500). Secondary antibodies used was donkey anti-rabbit RRX (Jackson ImmunoResearch Labs, 711295152). Fixed samples were imaged on a Zeiss 700 confocal microscope with a ×40 oil objective. Serial optic sections were obtained from whole-mount larval brains with a typical resolution of 0.09 μm × 0.09 μm × 0.47 μm.

For live imaging experiments, data were collected during a 4 h period in the subjective day (ZT1-ZT5). Larval brain explants were carefully separated from the rest of the larval tissue without damaging the brain lobes and then transferred into an external saline solution (120 mM NaCl, 4 mM MgCl2, 3 mM KCl, 10 mM NaHCO3, 10 mM Glucose, 10 mM Sucrose, 5 mM TES, 10 mM HEPES, 2 mM Ca2+, PH 7.2) and maintained in a chamber between the slide and cover-glass during the recording sessions. Time-lapse live imaging was performed on a Zeiss LSM 780 confocal microscope equipped with a Coherent Vision II multiphoton laser tuned to 920 nm. The samples were imaged with a ×40 water objective with typical x-y-z resolution at 0.11 μm × 0.11 μm × 0.25 μm; images series were taken at 1 min per Z-stack for a total of 10 min. The samples were excluded if the dendrite morphology is abnormal (e.g., disturbed or stretched during dissection and mounting) or if severe drifting occurred during imaging session that was uncorrectable by the subsequent image processing.

**Image processing and analysis.** 4D imaging data sets from the live imaging experiments were subjected to drift correction and deconvolution by Huygens Professional (Scientific Volume Imaging). Processed images were then analyzed with the 3D visualization software Imaris (Bitplane). All dendrite dynamics data were from single-labeled LNvs except for the results presented in Fig. 6a, b and Fig. 7. To quantify dendrite dynamics, all branch tips are marked manually then tracked starting from the first frame of the time-lapse series with the Spots tracking module. The 3D coordinates of the branch tips are exported as csv files and analyzed using a custom-written MATLAB script and R scripts. Branches appearing after the first frame are considered newly appeared branches. The extensions and retractions of these newly appeared branches are counted towards the total extension and retraction events. The extension and retraction are defined by a movement of the branch tips that travels more than 0.3 μm in distance in 1 min. Continuous extensions/retractions captured in consecutive frames are counted as one extension. Extensions/retractions captured in nonconsecutive frames are counted as separate events.

To quantify dendrite volume and the number of synaptic contacts, Z stacks of confocal images of dendritic arbors labeled by GFP were reconstructed in 3D using the Surface module of Imaris. The volume encompassed by individual reconstructed surfaces was then calculated and summed up to be the total volume of the dendritic tree. The same Surface reconstruction method was also applied to Extra-Fas2:YFP quantification.

Rh5,6-Brp:mCherry puncta were detected using the Spots module of Imaris with a threshold for the average diameter set at 0.6 μm[32]. The distance between the center of individual Brp-Cherry cluster and the reconstructed LNv dendrite surface was measured by the Spots Close To Surface XTension. Brp clusters contacting LNv dendrites, with a distance < 0.3 μm, were recognized as synaptic contacts between BN and LNv.

The investigators were blinded to the group allocation when quantifying data for the development or experience regulated dendrite dynamics, dendrite volume, and synaptic contact.

**Calcium imaging.** Late 3rd instar larvae expressing Pdf-Gal4 driving UAS-GCaMP6s were used for calcium imaging experiments. The procedures for sample preparation and two-photon imaging setup were the same as the live imaging experiments with some modifications. GCaMP6s signals were collected at 100 ms per frame for 2000 frames during each imaging session with the optic resolution at 256 × 90 pixels. Light stimulations of 100 ms duration were delivered by a 561 nm confocal laser controlled by the photo bleaching program. GCaMP6s signals at the axonal terminal region of LNvs were recorded and analyzed. Average GCaMP6s signals of 20 frames before light stimulation was taken as F0, and ΔF/F0 was calculated for each data point. The light stimulation was delivered three times for each sample with 50 s intervals. The mean of the 2nd and 3rd responses for each sample were used for quantifications. The sample number n represents number of individual animals subjected to the optical recordings.

**Statistical analysis.** Statistical analyses were performed using GraphPad Prism. Two-tailed unpaired Student's t-test was used to compare data in two groups with equal or unequal sample numbers. For data containing multiple groups, one-way ANOVA was used with follow-up Dunnett's (every group compared to Control) or Tukey's (every group compared to every other group) multiple comparisons test. Most quantitative data are presented as box plot overlaid with dot plot; box plot shows the median (center line in box), interquartile ranges (top and bottom of box, 25–75%), and total ranges (whiskers, 0–100%); dot plot displays individual data points. Some data are presented as histogram, bar chart, or line graph showing the mean ± SEM as indicated. $P \geq 0.05$ was considered not significant (ns); $*P < 0.05$, $**P < 0.01$, $***P < 0.001$.

**Code availability.** Custom MATLAB scripts and R scripts are used to speed up data analysis. They are available from the corresponding author upon reasonable request.

**Data availability.** All data supporting the findings in this study are available from the corresponding author upon reasonable request.

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

## Acknowledgements

This work is supported by the Intramural Research Program of NINDS. We thank Gabrielle Boulianne and Andrew Zelhof for anti-Amph antibodies and fly lines; Benjamin White, Chi-Hon Lee, Mark Stopfer, Sijun Zhu, Peter Soba, and Anna Kim for helpful discussions and comments on manuscripts.

## Author contributions

C.S. and Q.Y. designed the experiments, C.S. and Q.Y. collected live imaging data and performed data analysis with U.J., M.G., and J.Y.. C.L., M.G., and C.S. collected fixed imaging data and performed the data analysis. B.Q. contributed to the development of 4D tracing method and C.S. and Q.Y. wrote the manuscript.

## Additional information

**Competing interests:** The authors declare no competing interests.

