## [Peer Review File · Nature Communications]

Reviewers' comments:

Reviewer #1 (Remarks to the Author):

In this work Sheng and colleagues perform an elegant and detailed live imaging analysis of dendrite maturation using the *Drosophila* CNS LNV neurons as a model. They find that an early period of heightened filopodial dynamics correlates with synaptogenesis and this is strongly influenced by visual experience. Finally, using a small scale RNAi screen they find that Amphiphysin regulates several aspects of dendrite dynamics and maturation likely through a mechanism involving Fas2.

The analysis is careful and the paper well written and very interesting. While I found the wild type and experience-dependent characterization original and convincing, the mechanistic analysis is somewhat less complete and I would interpret that more carefully, and perhaps somewhat differently. This concern notwithstanding, the paper remains solid and very much worthy of publication in *Nature Communications*.

Below are a few comments that the authors may wish to consider:

1- The conclusion on page 9 that filopodial dynamics promote synaptogenesis is too strongly stated, as from pure observation this remains a correlation. It is fair to say that the two are coupled and related, but it is difficult to be certain about the nature of the causal link at this point.

2- What is the effect of DD conditions on branch dynamics at different stages and on synapse formation?

3- If I understood correctly, dendrite volume was not one of the parameters that seemed to change during development and to correlate with synapse formation. Yet, it is influenced by experience. How do the authors interpret this?

4- I wonder if the Amphiphysin loss of function and rescue data would be better interpreted as it being required for the coupling between dynamics and synaptogenesis.

5- I am not sure how essential the Fas2 data are for the major message of the paper.

Reviewer #2 (Remarks to the Author):

This manuscript focuses on identifying the 'role of filopodia motility in dendrite morphogenesis and experience-dependent plasticity', using a visual circuit in *Drosophila* larvae. Time lapse imaging of filopodial movements and dendrite morphology across development, and under different sensory stimulation paradigms were undertaken. The main conclusions are: (i) that there are more dynamic filopodia during early periods of synaptogenesis, and dendrites become more stable with maturation (as in other model systems), (ii) enhanced light exposure reduced exploratory filopodia, but caused more rapid

synaptogenesis and dendritic maturation, and (iii) results of a screen and RNAi experiments identified Amphiphysin as a cell autonomous regulator of dendrite dynamics through modulation of synaptic transmission. In general, this is an interesting and important topic in the developmental neuroscience field, and the authors have used state-of-the-art approaches to further test the importance of dynamic filopodia in synapse formation. My main concern is that it is unclear how the authors reached some of their major conclusions. In part, this may be the style of writing, where many sentences are broadly phrased, making it difficult to really understand the points that may be intended. Thus some conclusions appear overstated, beyond what the data support.

Specific comments:

1. Throughout the text, there are many sentences that are difficult to grasp: For example, in the Abstract – "dynamic filopodia is the key cellular target for homeostatic regulation of dendritic development". Why 'homeostatic'? This is referred to in many places, but I am not sure why the authors think that activity-dependent regulation of filopodial dynamics reflects a homeostatic mechanism? Indeed, the title "homeostatic structural plasticity targets dynamic filopodia in regulating dendrite maturation and synaptogenesis" does not clearly represent the findings – perhaps the writing is just confusing, but it will be important to state explicitly what is meant by 'homeostasis' in the current work.

2. The conclusion that Amph is "a regulator for dendrite dynamics and a new molecular component of postsynaptic compartment organization in the fly CNS" is misleading because Amph clearly has presynaptic effects as evidenced by the Amph^{-/-} observations, and because of the failure to completely rescue by overexpressing Amph in the LNV neurons in the mutant.

3. Support for the synaptotrophic hypothesis seems a bit tenuous - in the LL condition (Fig. 4e), it is argued that synapse number is down at 72 h but dendrite volume is not yet different, compared to LD conditions. But at 48 h, dendrite volume is already significantly lower between LL and LD, and synapse number is not different.

4. Figure 1: Distinction between an "extension" event (green) and a "newly appeared" (yellow) event is unclear. Please clarify in the methods section.

5. One cannot tell from the bruchpilot (Brp) staining shown in Fig. 3 where the synaptic contacts are. Higher magnification of the staining is needed to judge how well the CD8:GFP overlaps with the Brp:mCherry signal (is there sufficient resolution to identify these as synapses?). Same problem with Fig. 6a. Also, yellow puncta on a gray mask are difficult to resolve. See also Supplementary Figure 3 comment below.

6. Figure 4b: not clear what age is displayed in the graphs - is it 120hr?

7. Figure 5: The plots in this figure show that the number of synaptic contacts has not changed in the Amph RNAi condition, but it would be good to see images of the presynaptic Brp puncta in control vs RNAi condition. Perhaps a supplementary figure can be

amended to include this information. The authors show the Brp puncta distribution for the *Amph*^{-/-} condition but not for the RNAi experiments.

8. Figure 6: Increase in dendritic dynamics in the *Amph*^{-/-} as shown by the plot in C is much less severe than the increase of dynamics after *Amph* RNAi treatment. How do we reconcile the differences between the RNAi and the mutant observations? What is the distribution pattern of *Amph*? Could it be expressed by the BO neurons themselves and that is why the number of presynapses on the axon terminals is reduced in the *Amph* loss of function mutants? I may have missed this, but what was the duration of imaging in the mutant vs the RNAi experiments?

9. Figure 7: The authors claim that the Fas2-YFP signal on LNV dendrites represents the synaptic distribution pattern on these dendrites and that this distribution is altered (reduced) in *Amph* RNAi knockdown conditions. It would be informative to confirm that Fas2 on LNV dendrites is opposite Brp presynaptic puncta in normal and *Amph* knockdown so as to provide confidence that the Fas2 changes are synaptic.

10. Suppl Figure 2: Why are there more stable branches at the 96 hr time point compared to the later 120 hr time point? Are any of the differences significant in this plot?

11. Suppl Figure 3 and others: Definition of "synaptic puncta" is unclear. The distance from the center of the Brp punctum to the surface of the LNV dendrites is measured and if that is within 0.3 micrometers, the punctum is counted as synaptic. The number of synaptic contacts may be over-estimated with the current analysis. First, how large are these Brp punctum, and how are they identified? Given the resolution of light, one would imagine that there should be pixel overlap between the Brp punctum and the dendrite signal. Rather than measure the center of the punctum to the dendrite, one should just look for colocalization of the two channels; there should be overlap. Also, controls are needed; for example, flipping the image of one channel (just within the region with the dendrites and puncta) before masking the Brp signal and looking at colocalization.

12. Suppl Figure 4: Statistics needs to be provided - also please indicate what time point.

Reviewer #3 (Remarks to the Author):

18-03 Nat Comm

This study by Sheng et al from the Yuan lab uses the well-studied *Drosophila* LNV circadian neurons to investigate the molecular basis of dendrite dynamics and experience-dependent plasticity. In this case, the sensory stimulus is light as the LNVs are innervated by the larval visual system and respond to light inputs. The authors present a careful and detailed analysis of dendrite dynamics in the LNVs and identify that Amphiphysin is required for LNVs to reduce dendrite dynamics in response to constant light. The quality of the data is high and well-quantified but I have some concerns with how new some of the work is and with

the importance of the findings.

1. Figures 1-4 seem to be a more detailed characterization of the dendrite morphology changes originally described in Yuan et al, 2011. At the end of these 4 figures, the authors state that this is a "newly established system". While I agree that the live-imaging is a new addition to this current study (but see #2 below), this does not seem like a new system to me. Perhaps the authors need to summarize what they have learned from Figures 1-4 that we did not know already.

2. The amount of filopodia movements during the imaging are very interesting but also surprising, at least to me. Can the authors exclude that the dendritic movements are due to the light from the microscope? I think the answer is probably no since there are large differences between 2nd and 3rd instar larvae, but this is worth mentioning.

3. Given the effects of light on LNV dendrites, I was surprised that the authors only use developmental timing (hours after egg laying) and do not mention the time in an LD cycle that larvae were taken from. Were the larvae maintained in LD cycles and what time of day were they taken from? It may be that the time in an LD cycle does not affect LNV dendrites, but this would also be important to know.

4. There are several studies on plasticity of the projections of the adult LNvs. It is interesting that 2 of the genes that the authors mention in the beginning of Figure 5 (Fas2 and Rho1) have been shown to affect adult LNV plasticity. So it is probably worth mentioning these studies from the Rosbash and Blau labs respectively. Also the authors decided "not to show" the Fas2 data. I thought all data needed to be shown in these days of supplemental data. Similarly, shouldn't all of the screen data be included? And did the authors revisit the genes that Yuan et al identified in the 2011 paper in Science?

5. For Figure 5, it would be important to know if RNAi to Amph also increases dendrite dynamics in LD cycles. If so, then this would indicate that Amph non-specifically increases dendrite dynamics and that would make the conclusions of the paper less interesting, in my opinion. If the effect of Amph RNAi on dendrite dynamics is restricted to LL, then that would support the idea that light regulates dendrite dynamics via Amph, which would be a more interesting conclusion.

6. In the graph with the # of synaptic contacts in Figure 6b, it looks like the Amph mutants consist of two groups: one centered around 40 synaptic contacts and another around 75. Is the mutation differentially penetrant? A side note, I also had a difficult time in finding Figure 6c at first.

7. The authors should speculate on why Uas-Amph did not give a phenotype. I agree that the RNAi data are more important since they reduce Amph function, but it is good to address data such as the Uas-Amph that do not give an effect.

8. How was the light applied in Figure 7a?

9. Given the conservation of biology, why do the authors find the degree of similarity “surprising” between LNvs and developing vertebrate neurons?

10. I found the nomenclature Rh5,6>Brp confusing. The authors state that this is a direct fusion, but then use the same > symbol to indicate Gal4/UAS.

Response to the reviewers' and editor's comments

We greatly appreciate the reviewers' evaluations of our work as well as the editor's suggestions to address specific technical points and edit the text to appropriately state claims. The reviewers did not raise major concerns with our experimental designs or main conclusions but offered many constructive suggestions. We conducted new experiments to address the reviewers' questions and provide additional information to improve clarity and accuracy. We believe that we have addressed all comments and our manuscript has been significantly improved. We hope that the editor and reviewers will find our revised paper now suitable for Nature Communication. The following is a summary of the major changes made and our point-by-point response to reviewers' comments.

Major changes included in the revised manuscript:

1. We provided images with higher magnification for the 3D reconstruction of the BN-LNv synaptic contacts in Figure 3 and 6, as suggested by Reviewer 2. We also added a new supplementary figure containing high resolution 3D reconstruction images of the Rh5,6-Brp:mCherry puncta contacting LNv dendrites (new Fig. S4).
2. We added new results to show the dendritic distribution of extra-Fas2:YFP in relation to the Rh5,6-Brp puncta and the effect of Amph knockdown on the distribution, as suggested by Reviewer 2 (Fig. 7f-h).
3. We added new results describing dendrite dynamics and synaptogenesis phenotypes associated with Fas2, as suggested by Reviewer 3 (Fig. S6).
4. We added new results to show the broad expression pattern of Amph in the larval CNS and LNvs (Fig. S10), to address the questions raised by Reviewer 2 and 3.
5. We added a supplementary Excel file containing the list of the 134 genes screened for dynamics phenotypes.
6. We added the technical information related to experimental setup and quantification in the Methods section, as suggested by Reviewers 2 and 3
7. **[redacted]**
8. Following the reviewers' and editor's suggestions, we modified the Introduction to reflect the connections between this work and our previous studies on the structural homeostasis during LNv dendrite development. In addition, we modified the statement in the Results section to appropriately state our findings and added a section in the Discussion to describe phenotypes obtained from the Amph mutant and KD conditions.

Point-by-point response to reviewers' comments

Reviewer #1 (Remarks to the Author):

In this work Sheng and colleagues perform an elegant and detailed live imaging analysis of dendrite maturation using the Drosophila CNS LNV neurons as a model. They find that an early period of heightened filopodial dynamics correlates with synaptogenesis and this is strongly influenced by visual experience. Finally, using a small scale RNAi screen they find that Amphiphysin regulates several aspects of dendrite dynamics and maturation likely through a mechanism involving Fas2.

The analysis is careful and the paper well written and very interesting. While I found the wild type and experience-dependent characterization original and convincing, the mechanistic analysis is somewhat less complete and I would interpret that more carefully, and perhaps somewhat differently. This concern notwithstanding, the paper remains solid and very much worthy of publication in Nature Communications.

Below are a few comments that the authors may wish to consider:

1- The conclusion on page 9 that filopodial dynamics promote synaptogenesis is too strongly stated, as from pure observation this remains a correlation. It is fair to say that the two are coupled and related, but it is difficult to be certain about the nature of the causal link at this point.

Response:

We agree that the current data support a correlation, not a causal link. We modified the statement on P9 as below:

These observations support our model in which young dendritic arbors contain a high percentage of dynamic filopodia that correlate with synaptogenesis. As neurons mature, dendrites make the transition into a stable state that supports continued growth.

2- What is the effect of DD conditions on branch dynamics at different stages and on synapse formation?

Response:

DD conditions produced an elevated dendrite dynamic at 120hr AEL (**Fig. 4a, b**). In addition, our previous study demonstrated that dendrite volume and synapse number both increase in DD conditions (Yuan. et al, 2011). These observations are consistent with the notion that DD conditions extend the synaptogenesis period through increased dendrite dynamics. However, at the early stages of development, the prevalence of dynamic dendritic filopodia are extremely high. It is difficult to accurately quantify a further increase of dendrite dynamics generated by manipulations of conditions or genotypes. Therefore, in this study, we focused our efforts on analyzing LL conditions that reduce dendrite dynamics.

[redacted]

3- If I understood correctly, dendrite volume was not one of the parameters that seemed to change during development and to correlate with synapse formation. Yet, it is influenced by experience. How do the authors interpret this?

Response:

The LNV dendrite volume increases during development and plateaus at 96 hr in LD conditions, about 24 hours after synapse saturation (**Fig. 3c**). In LL conditions, the expansion of dendrite volume plateaus at 72 hr, 24 hours in advance compared to the LD conditions, correlating with the premature termination of the synaptogenesis period (**Fig. 4d**). Therefore, although dendrite volume expansion does not exhibit a direct temporal correlation with the synapse formation, it is modified by the number of synapses formed during the dynamic phase of the dendrite development. Alterations in visual experience, such as the LL condition, produce changes in dendrite dynamics and synaptogenesis, indirectly influencing dendrite growth. We described these observations in the results (P9-10) and discussion (P17-19).

4- I wonder if the Amphiphysin loss of function and rescue data would be better interpreted as it being required for the coupling between dynamics and synaptogenesis.

Response:

We appreciate the suggestion from the reviewer. In fact, a decoupling between dynamics regulation and synaptogenesis would agree with the phenotypes we observed in LNvs with an Amph knock-down, which show an increase in dynamics without deficits in synaptogenesis. However, the results from the loss-of-function mutants of Amph and the rescue experiments indicate additional roles for Amph in synapse formation and organization. We included a paragraph in the discussion to describe these proposed functions of Amph (P18).

5- I am not sure how essential the Fas2 data are for the major message of the paper.

Response:

The connection between Amph and Fas2, a well-established synaptic adhesion molecule, support the role of Amph in organizing the postsynaptic compartment on dendrites. We demonstrate the impact of the dendritic distribution of Fas2 by Amph knock-down in LNvs (**Fig. 7 d, e**) and include additional evidence to support this result in the revision (**Fig. 7f-h**).

Reviewer #2 (Remarks to the Author):

This manuscript focuses on identifying the ‘role of filopodia motility in dendrite morphogenesis and experience-dependent plasticity’, using a visual circuit in Drosophila larvae. Time lapse imaging of filopodial movements and dendrite morphology across development, and under different sensory stimulation paradigms were undertaken. The main conclusions are: (i) that there are more dynamic filopodia during early periods of synaptogenesis, and dendrites become more stable with maturation (as in other model systems), (ii) enhanced light exposure reduced exploratory filopodia, but caused more rapid synaptogenesis and dendritic maturation, and (iii) results of a screen and RNAi experiments identified Amphiphysin as a cell autonomous regulator of dendrite dynamics through modulation of synaptic transmission. In general, this is an interesting and important topic in the developmental neuroscience field, and the authors have used state-of-the-art approaches to further test the importance of dynamic filopodia in synapse formation. My main concern is that it is unclear how the authors reached some of their major conclusions. In part, this may be the style of writing, where many sentences are broadly phrased, making it difficult to really understand the points that may be intended. Thus some conclusions appear overstated, beyond what the data support.

Specific comments:

1. Throughout the text, there are many sentences that are difficult to grasp: For example, in the Abstract – " dynamic filopodia is the key cellular target for homeostatic regulation of dendritic

development". Why 'homeostatic'? This is referred to in many places, but I am not sure why the authors think that activity-dependent regulation of filopodial dynamics reflects a homeostatic mechanism? Indeed, the title "homeostatic structural plasticity targets dynamic filopodia in regulating dendrite maturation and synaptogenesis" does not clearly represent the findings – perhaps the writing is just confusing, but it will be important to state explicitly what is meant by 'homeostasis' in the current work.

Response:

We appreciate the reviewer's comment and modified the introduction to better introduce the homeostatic structural plasticity we observed during LNV dendrite development (P4).

In our previous study (Yuan et al. 2011), we found that the amount of light exposure or the level of neuronal activity is inversely correlated with the LNV dendrite size. We believe this is an example of the compensatory homeostatic mechanism that regulates dendrite growth based on the input activity. In this study, using live imaging approaches combined with the temporal profiling of synaptogenesis and dendrite growth, we identified dendrite dynamics as the cellular substrate for this homeostatic regulation. Our results indicate that LL conditions promote dendrite maturation by reducing the prevalence of dynamic filopodia, which correlate to decreased synapse formation and reduced dendrite growth in the later stages. We believe this is the key finding of this study and therefore will keep the current title of the manuscript.

2. The conclusion that Amph is "a regulator for dendrite dynamics and a new molecular component of postsynaptic compartment organization in the fly CNS" is misleading because Amph clearly has presynaptic effects as evidenced by the Amph-/- observations, and because of the failure to completely rescue by overexpressing Amph in the LNV neurons in the mutant.

Response:

We agree that Amph has non-autonomous effects on LNV dendrite development and synaptogenesis. We included this statement in the Results section (P14). However, our results and previous studies also indicate *Drosophila* Amph as an important component for the postsynaptic compartment. It has been shown previously at the NMJ that the loss-of-function mutation of Amph does not cause functional deficits in presynaptic terminals, but rather it influences the localization of several postsynaptic proteins (P15). These findings are consistent with our observations in the larval CNS. Using a LNV specific knock-down approach, our study provided evidence for Amph's autonomous roles in regulating LNV dendrite dynamics, the dendritic distribution of Fas2 and the physiological responses of LNVs to light stimulation.

3. Support for the synaptotrophic hypothesis seems a bit tenuous - in the LL condition (Fig. 4e), it is argued that synapse number is down at 72 h but dendrite volume is not yet different, compared to LD conditions. But at 48 h, dendrite volume is already significantly lower between LL and LD, and synapse number is not different.

Response:

We agree with the reviewer that not all the data points support our model. Specifically, the dendrite volume of the LL sample at 48 hr AEL is significantly lower than the one in LD. However, in general, the temporal profiles of synapse formation and dendrite expansion are clearly modified by the LL condition, which also strongly influences the dendrite dynamics. Together, these results support a correlation between the state of dendrite dynamics and synaptogenesis. A modification of our language was also suggested by reviewer 1 in comment #1. We amended our statement in P10.

4. Figure 1: Distinction between an "extension" event (green) and a "newly appeared" (yellow) event is unclear. Please clarify in the methods section.

Response:

We added a paragraph in the methods section to clarify the definitions:

Branches appearing after the first frame are considered newly appeared branches. The extensions and retractions of these newly appeared branches are counted towards the total extension and retraction events. The extension and retraction are defined by a movement of the branch tips that travels more than 0.3 μm in distance in 1 min. Continuous extensions/retractions captured in consecutive frames are counted as one extension. Extensions/retractions captured in nonconsecutive frames are counted as separate events.

5. One cannot tell from the bruchpilot (Brp) staining shown in Fig. 3 where the synaptic contacts are. Higher magnification of the staining is needed to judge how well the CD8:GFP overlaps with the Brp:mCherry signal (is there sufficient resolution to identify these as synapses?). Same problem with Fig. 6a. Also, yellow puncta on a gray mask are difficult to resolve. See also Supplementary Figure 3 comment below.

Response:

We modified the figures according to the reviewer's suggestion. To clearly show the reconstructed synaptic contacts, we zoomed in on the dendritic region and increased the contrast of the images (see updated **Fig.3** and **Fig.6**, and new **Fig. S3, S6** and **S7**). The contacting spots are clearly visible now.

We also added an additional supplementary figure to provide high resolution 3D reconstruction images of Rh5,6-Brp:mCherry puncta contacting LNV dendrites (see **Fig. S4**). Please also see our response to comment #11.

6. Figure 4b: not clear what age is displayed in the graphs - is it 120hr?

Response:

Yes. We add the information in the figure legend.

7. Figure 5: The plots in this figure show that the number of synaptic contacts has not changed in the Amph RNAi condition, but it would be good to also see images of the presynaptic Brp puncta in control vs RNAi condition. Perhaps a supplementary figure can be amended to include this information. The authors show the Brp puncta distribution for the Amph^{-/-} condition but not for the RNAi experiments.

Response:

We added a supplementary figure (**Fig. S7**) to include that information.

8. Figure 6: Increase in dendritic dynamics in the Amph^{-/-} as shown by the plot in C is much less severe than the increase of dynamics after Amph RNAi treatment. How do we reconcile the differences between the RNAi and the mutant observations?

Response:

We were unable to perform single cell labeling in the mutant background. Therefore, the dendritic dynamic phenotype of Amph^{-/-} and its background control in **Fig. 6c** were quantified from larvae raised in LL. Compared to the LD condition, LNv dendrites in LL conditions have both reduced dendrite volume and dynamics, which are necessary for the quantification without single cell resolution. The experimental conditions were described in P13 and the Methods section.

What is the distribution pattern of Amph? Could it be expressed by the BO neurons themselves and that is why the number of presynapses on the axon terminals is reduced in the Amph loss of function mutants?

Response:

The severe reduction of Rh5,6-Brp puncta in Amph mutants suggests important functions of Amph in synaptogenesis, consistent with its wide distribution in the developing CNS as reported previously (P13). To characterize Amph's distribution on the LNv dendrite, we obtained the anti-Amph antibody for immunostaining and an Amph enhancer-Gal4 line to drive CD8:GFP expression. Results generated by both approaches confirmed the broad and diffuse distributions of Amph in the CNS. With thin optical sections, we observed Amph staining on LNv dendrites. In addition, there appears to be no clear overlap of Amph signal with Rh5,6-Brp puncta, suggesting Amph is not enriched in the presynaptic terminals of photoreceptor cells in the BO (See new **Fig. S10**). We include this in the Results section in P14.

I may have missed this, but what was the duration of imaging in the mutant vs the RNAi experiments?

Response:

All time-lapse live imaging sessions are 10 min long as described in the Methods.

9. Figure 7: The authors claim that the Fas2-YFP signal on LNV dendrites represents the synaptic distribution pattern on these dendrites and that this distribution is altered (reduced) in Amph RNAi knockdown conditions. It would be informative to confirm that Fas2 on LNV dendrites is opposite Brp presynaptic puncta in normal and Amph knockdown so as to provide confidence that the Fas2 changes are synaptic.

Response:

We performed the experiment suggested by the reviewer. The results are included in modified **Fig. 7f-h** and described in P16. 3D reconstructions of LNV dendrites expressing extra-Fas2:YFP and Rh5, 6-Brp:mCherry clearly illustrate the Fas2:YFP labeled dendrites in close contact with the Brp puncta. Moreover, Amph knockdown reduces these Brp-Fas2 contacts. This new result supports the function of Amph in regulating the synaptic distribution of Fas2.

10. Suppl Figure 2: Why are there more stable branches at the 96 hr time point compared to the later 120 hr time point? Are any of the differences significant in this plot?

Response:

There is no statistical difference between 96 hr and 120 hr. In terms of dendrite dynamics, 96 hr and 120 hr are very similar in all parameters.

11. Suppl Figure 3 and others: Definition of "synaptic puncta" is unclear. The distance from the center of the Brp punctum to the surface of the LNV dendrites is measured and if that is within 0.3 micrometers, the punctum is counted as synaptic. The number of synaptic contacts may be over-estimated with the current analysis. First, how large are these Brp punctum, and how are they identified? Given the resolution of light, one would imagine that there should be pixel overlap between the Brp punctum and the dendrite signal. Rather than measure the center of the punctum to the dendrite, one should just look for colocalization of the two channels; there should be overlap. Also, controls are needed; for example, flipping the image of one channel (just within the region with the dendrites and puncta) before masking the Brp signal and looking at colocalization.

Response:

As cited in the Methods section, the technique we used was developed by Liqun Luo's lab for quantifying putative synaptic contacts in *Drosophila* CNS (Mosca, T. and Luo, L., 2014). The average diameter of the Brp puncta is 0.6 μ m. Therefore, a distance of 0.3 μ m or less between Brp puncta and dendritic arbor indicates a potential contact. This threshold setting was further validated in our system by manual counting. As suggested by the reviewer, high resolution images of the spot reconstruction and contact spot detection using Imaris are provided in the new **Fig. S4**, which support the accuracy of this method.

We also tested several intensity-based quantification methods, including measuring the colocalization of the two channels, none of which generated consistent measurements. It is likely that the dense dendritic arbors are difficult to accurately quantify by intensity measurements. Our current method employs both 3D volume reconstruction and 3D spot counting, requires separate constructions of dendrite arbors and synaptic contacts for every sample, and is very labor intensive, but the results are consistent across independent experiments as demonstrated by many measurements shown in **Fig. 3-6** and **Fig. S6, S9**. Therefore, although we cannot completely exclude the possibility of a small overestimate of the synaptic contact number, we are confident that our current methods generate quantitative measurements of the putative synaptic contacts that are largely reflecting the real synapses.

12. Suppl Figure 4: Statistics needs to be provided - also please indicate what time point.

Response:

The statistics and the time point, 120hr, were added for that figure (now **Fig.S5**).

Reviewer #3 (Remarks to the Author):

18-03 Nat Comm

This study by Sheng et al from the Yuan lab uses the well-studied Drosophila LNV circadian neurons to investigate the molecular basis of dendrite dynamics and experience-dependent plasticity. In this case, the sensory stimulus is light as the LNVs are innervated by the larval visual system and respond to light inputs. The authors present a careful and detailed analysis of dendrite dynamics in the LNVs and identify that Amphiphysin is required for LNVs to reduce dendrite dynamics in response to constant light. The quality of the data is high and well-quantified but I have some concerns with how new some of the work is and with the importance of the findings.

1. Figures 1-4 seem to be a more detailed characterization of the dendrite morphology changes originally described in Yuan et al, 2011. At the end of these 4 figures, the authors state that this is a “newly established system”. While I agree that the live-imaging is a new addition to this current study (but see #2 below), this does not seem like a new system to me. Perhaps the authors need to summarize what they have learned from Figures 1-4 that we did not know already.

Response:

We thank the reviewer for recognizing our previous work. In fact, one of the main motivations of the current study is to identify the cellular mechanism underlying the homeostatic plasticity we previously reported (Yuan et al. 2011). And it is very satisfying that we are able to determine how elevated input activity modifies the size of the LNV dendrite through its effect on dendrite dynamics and synaptogenesis. During this process, we also established a new system to perform genetic studies on synapse-forming dendritic filopodia in the *Drosophila* CNS (**Fig. 1-2**),

identified distinct temporal profiles and regulations of dendrite growth and dendrite dynamics (Fig. 3) and found the effect of experience in regulating the prevalence of dendrite filopodia and dendrite maturation (Fig. 4). Based on the reviewer's suggestions, we modified the introduction and discussion to better represent the significance of this current work, see P4 and P17.

2. The amount of filopodia movements during the imaging are very interesting but also surprising, at least to me. Can the authors exclude that the dendritic movements are due to the light from the microscope? I think the answer is probably no since there are large differences between 2nd and 3rd instar larvae, but this is worth mentioning.

Response:

The dynamic behaviors of the dendrite branches in LNvs are indeed very striking. At the time we established the system, to exclude the possibility of light induced acute changes in dendrite dynamics, we performed comparisons between the preparation with the BO intact or removed at 120hr AEL. The quantification showed a similar amount of dynamics in both types of preparations and suggests that the BO-mediated phototransduction is not interfering with the ongoing dendrite dynamics. The results are shown below. In addition, our time-lapse live imaging experiments were performed using 2-photon microscopy with the 920 nm laser. The phototoxicity and light stimulation were kept to a minimum under this condition.

3. Given the effects of light on LNv dendrites, I was surprised that the authors only use developmental timing (hours after egg laying) and do not mention the time in an LD cycle that larvae were taken from. Were the larvae maintained in LD cycles and what time of day were they taken from? It may be that the time in an LD cycle does not affect LNv dendrites, but this would also be important to know.

Response:

Although our previous study indicated that the LNV dendritic plasticity was not affected by circadian timing, we maintain all our cultures in circadian controlled incubators and all live imaging experiments are performed during a 4 hour slot on the subjective day (ZT1-ZT5). This is now described in the Methods section.

4. There are several studies on plasticity of the projections of the adult LNvs. It is interesting that 2 of the genes that the authors mention in the beginning of Figure 5 (Fas2 and Rho1) have been shown to affect adult LNV plasticity. So it is probably worth mentioning these studies from the Rosbash and Blau labs respectively.

Response:

We referenced the studies from Rosbash and Blau labs as suggested by the reviewer in P12.

Also the authors decided “not to show” the Fas2 data. I thought all data needed to be shown in these days of supplemental data. Similarly, shouldn’t all of the screen data be included? And did the authors revisit the genes that Yuan et al identified in the 2011 paper in Science?

The live imaging screens were performed by visual inspection only and had no quantifications. Therefore, we did not include the results in our initial submission. Here, we added a supplementary file listing the genes screened in our live imaging analyses and indicated the candidate molecules associated with the altered dendrite dynamics. The screen was focused on cytoskeleton-associated proteins, motor proteins and synapse components, a different set of genes from our previous publication.

Among the candidate genes, we validated the dynamic phenotypes of Amph and Fas2 using single cell labeling. As suggested by the reviewer, we included the results related to Fas2 in the new **Fig. S6**. To test the loss-of-function effect of Fas2 with consistency, we expressed a dominant-negative form of Fas2 (Fas2 Δ 3) in LNvs, which elevated dendrite dynamics while reducing the dendrite volume and synaptic contacts. This is described in the Results on P12.

5. For Figure 5, it would be important to know if RNAi to Amph also increases dendrite dynamics in LD cycles. If so, then this would indicate that Amph non-specifically increases dendrite dynamics and that would make the conclusions of the paper less interesting, in my opinion. If the effect of Amph RNAi on dendrite dynamics is restricted to LL, then that would support the idea that light regulates dendrite dynamics via Amph, which would be a more interesting conclusion.

Response:

In **Figure 5**, the quantitative measurement of dendrite dynamics phenotype of an Amph knock-down were collected in LD conditions. Therefore, the effect of Amph RNAi on dendrite dynamics is not restricted to LL. However, we think this does not weaken the role of Amph in regulating dendrite dynamics, which are determined by both developmental mechanisms and neuronal activity. Amph is likely a part of the developmental program that controls the prevalence of dynamic filopodia, suggested by its upregulated transcript level in late stages of

larval development (**Fig. S8**). Additionally, its level or distribution is potentially modified by LL to promote dendrite maturation. We include this statement in the discussion in P18.

6. In the graph with the # of synaptic contacts in Figure 6b, it looks like the Amph mutants consist of two groups: one centered around 40 synaptic contacts and another around 75. Is the mutation differentially penetrant? A side note, I also had a difficult time in finding Figure 6c at first.

Response:

Both Amph mutants we tested are complete loss-of-function (**Fig. 6, Fig. S8**). We also validated their genotypes with PCR. Therefore, it is unlikely that there is an issue of penetrance. We modified the color scheme of **Figure 6c** to make it distinguishable from the rest of the panels.

7. The authors should speculate on why Uas-Amph did not give a phenotype. I agree that the RNAi data are more important since they reduce Amph function, but it is good to address data such as the Uas-Amph that do not give an effect.

Response:

We added the following statement in P14:

Although reducing Amph leads to elevated dendrite dynamics, increasing its expression alone may not be sufficient to promote stabilization of the dendritic arbors. It is also possible that the Pdf-Gal4 driven expression of the Amph transgene did not achieve either the necessary protein level or the proper localization required for the dendrite stabilization phenotype.

8. How was the light applied in Figure 7a?

Response:

As described in the Methods section, the 100ms light stimulations were delivered using a photobleaching protocol controlled by the Zen software. The GCaMP signals were collected using a 920 nm 2-photon laser.

9. Given the conservation of biology, why do the authors find the degree of similarity “surprising” between LNvs and developing vertebrate neurons?

Response:

Our study establishes the first model for studying dynamic filopodia on the synapse forming dendrites of a *Drosophila* CNS neuron. There has no previous report of highly motile dendritic branches in the fly central synapse. Therefore, it is very exciting and surprising for us, after all the quantitative measurements, to discover that these dynamic filopodia behave in very much the same way as the ones found in chick RGC and Xenopus optic tectal neurons. These similarities suggest a common underlying cellular machinery and strengthen the argument that the molecular

understanding obtained in *Drosophila* study can be applied to the vertebrate system. To make sure we convey this message appropriately, we modified the statement in the discussion on P17.

There is a large degree of similarity between the dendritic filopodia of *Drosophila* larval LNvs and those in developing vertebrate neurons.

10. I found the nomenclature Rh5,6>Brp confusing. The authors state that this is a direct fusion, but then use the same > symbol to indicate Gal4/UAS.

Response:

We changed the nomenclature as suggested by the reviewer. All Gal4/UAS is now '>', and direct fusion is '-'. Fusion between protein and fluorophore is separated by ':'.

REVIEWERS' COMMENTS:

Reviewer #1 (Remarks to the Author):

The authors have addressed my concerns to a satisfactory level. I have no further comments.

Reviewer #2 (Remarks to the Author):

The authors have made substantial revisions, which have improved the manuscript in many ways. I have a few remaining issues, which I hope the authors will take into consideration. I still am not understanding why the authors think that the changes they see with sensory experience are 'homeostatic' events. Increasing or decreasing sensory experience has an inverse relationship to dendrite motility and synaptogenesis – the endpoint is not the same i.e. homeostatic mechanisms are usually engaged to ensure that the 'parameters' (e.g. synapse density, dendritic arbor size) are maintained despite changes in the 'external' environment. I think the results show that in this drosophila circuit, there cellular and molecular mechanisms that regulate synaptogenesis/dendrite morphogenesis etc. Adding the word 'homeostatic' really confuses the reader.

Other points:

1. The authors have improved significantly the illustration and quantification of the 'synapses' by showing higher magnification images. However, I don't think that the 'dots' demarcating the locations of presynaptic (brp) puncta (e.g. Figures 6,7) are meaningful because the expression of this presynaptic marker does not look very punctate to me. Imaris can certainly mark locations of 'puncta' given a set of criteria but I think it would not represent the data very well to have these dots mark presynaptic sites (which would imply these are where synaptic sites are localized). Generally, it appears that overlap of the two channels (brp and dendrite label) are where putative synapses could be. I think this is fine, but I would suggest removing the brp 'spots' from the figures.
2. Figure S10: It might be the pdf, but I could not tell where the 'pink' outline is.
3. The anti-Amph staining appears weak and noisy (Figure S10c,d ; there appears very little endogenous Amph signal in the LN dendrites. Perhaps this is why the rescue (expressing Amph in LN neurons in the Amph nulls) do not affect the loss of synapses and decrease in dendrite volume? Maybe the authors can clarify this point.

Reviewer #3 (Remarks to the Author):

The authors have responded well to my comments.

Point-by-point response to reviewers' comments

Reviewer #1 (Remarks to the Author):

The authors have addressed my concerns to a satisfactory level. I have no further comments.

Reviewer #2 (Remarks to the Author):

The authors have made substantial revisions, which have improved the manuscript in many ways. I have a few remaining issues, which I hope the authors will take into consideration. I still am not understanding why the authors think that the changes they see with sensory experience are 'homeostatic' events. Increasing or decreasing sensory experience has an inverse relationship to dendrite motility and synaptogenesis – the endpoint is not the same i.e. homeostatic mechanisms are usually engaged to ensure that the 'parameters' (e.g. synapse density, dendritic arbor size) are maintained despite changes in the 'external' environment. I think the results show that in this drosophila circuit, there cellular and molecular mechanisms that regulate synaptogenesis/dendrite morphogenesis etc. Adding the word 'homeostatic' really confuses the reader.

Response:

We appreciate the reviewer's comment. Although we believe the structural plasticity we observed in LNV dendrites is an example of the compensatory homeostatic mechanism that regulates dendrite growth based on the input activity, we understand that there may be confusion or misunderstanding raised by using the term "homeostatic plasticity". Therefore, as suggested by the reviewer and editor, we removed the term from the title, abstract and the main text except in the last sentences of the introduction and discussion (P5 and P19), where the use of the term is referring to the general phenomenon of structural homeostasis, but not the current study.

Other points:

1. The authors have improved significantly the illustration and quantification of the 'synapses' by showing higher magnification images. However, I don't think that the 'dots' demarcating the locations of presynaptic (brp) puncta (e.g. Figures 6,7) are meaningful because the expression of this presynaptic marker does not look very punctate to me. I can certainly mark locations of 'puncta' given a set of criteria but I think it would not represent the data very well to have these dots mark presynaptic sites (which would imply these are where synaptic sites are localized). Generally, it appears that overlap of the two channels (brp and dendrite label) are where putative synapses could be. I think this is fine, but I would suggest removing the brp 'spots' from the figures.

Response:

In our previous response to reviewer's comments (comments #5 and #11 from Reviewer 2), we provided detailed justifications and additional supplementary figures for methods we used in

quantifying synaptic contacts. Reviewer 2 now agrees that the synaptic contacts can be illustrated by the overlap of Brp and the dendrites, but still has issues about the Brp-puncta presented in Figure 7 and 8 (Figure 6 and 7 in the previous version). Since the data presented in Figure 7 is important for describing the synaptogenesis phenotypes of the Amph mutants and is similar in style and content with Figure 3, supplementary Figure 3, 6 and 7, we will not modify the content of Figure 7. However, to confirm the effect of Amph knock-down on the synaptic distribution of Fas2-YFP, we performed the experiments presented in Figure 8 f-h, where the Brp-puncta reconstruction are now removed as suggested by the reviewer.

2. Figure S10: It might be the pdf, but I could not tell where the 'pink' outline is.

Response:

We thank the reviewer for pointing out our error. We included the correct legend in the revised supplementary figure 10:

-the diffused distribution of Amph in the LON and in the soma and dendritic region of LNvs (dashed gray outlines).

3. The anti-Amph staining appears weak and noisy (Figure S10c,d ; there appears very little endogenous Amph signal in the LN dendrites. Perhaps this is why the rescue (expressing Amph in LN neurons in the Amph nulls) do not affect the loss of synapses and decrease in dendrite volume? Maybe the authors can clarify this point.

Response:

We agree that the images generated by the anti-Amph staining are noisy. The Amph protein associates with many membranous structures and has a wide distribution in larval brain tissues, likely causing the diffuse pattern and noisy appearance of the Amph signals. However, we do think it is clear that the level of Amph on the LNv dendritic region (large dashed ovals) is not low comparing to its level in the soma (small dashed ovals). The main point of this figure is to illustrate the expression of Amph protein in LNvs and that the Amph signal is not overlapping with the presynaptic terminals marked by Brp-puncta. We believe the images support these conclusions.

Reviewer #3 (Remarks to the Author):

The authors have responded well to my comments.